# MASPOB: Bandit-Based Prompt Optimization for Multi-Agent Systems with Graph Neural Networks

**Zhi Hong** [* 1 2]   **Qian Zhang** [* 1 3]   **Jiahang Sun** [1]   **Zhiwei Shang** [1]   **Mingze Kong** [1]   **Xiangyi Wang** [1]   **Yao Shu** [4]
**Zhongxiang Dai** [1]

## Abstract

Large Language Models (LLMs) have achieved substantial success in real-world applications, particularly as the cognitive backbone of Multi-Agent Systems (MAS) for orchestrating complex workflows. Since many deployments preclude workflow modifications while MAS performance is highly prompt-sensitive, prompt optimization becomes a critical strategy for improvement. However, real-world prompt optimization for MAS is impeded by three key challenges: (1) the need of sample efficiency due to prohibitive evaluation costs, (2) topology-induced coupling among prompts, and (3) the combinatorial explosion of the search space. To address these challenges, we introduce **MASPOB** (**M**ulti-**A**gent **S**ystem **P**rompt **O**ptimization via **B**andits), a novel sample-efficient framework based on bandits. By leveraging Upper Confidence Bound (UCB) to quantify uncertainty, the bandit framework balances exploration and exploitation, maximizing gains within a strictly limited budget. To handle topology-induced coupling, MASPOB integrates Graph Neural Networks (GNNs) to capture structural priors, learning topology-aware representations of prompt semantics. Furthermore, it employs coordinate ascent to decompose the optimization into univariate sub-problems, reducing search complexity from exponential to linear. Extensive experiments across diverse benchmarks demonstrate that MASPOB achieves state-of-the-art performance, consistently outperforming existing baselines. Our code is available at https://github.com/HZ1008/MASPOB.

---
[*]Equal contribution  [1]The Chinese University of Hong Kong, Shenzhen [2]South China University of Technology [3]Ritsumeikan University [4]The Hong Kong University of Science and Technology (Guangzhou). Correspondence to: Yao Shu <yaoshu@hkust-gz.edu.cn>, Zhongxiang Dai <daizhongxiang@cuhk.edu.cn>.

*Proceedings of the 43rd International Conference on Machine Learning*, Seoul, South Korea. PMLR 306, 2026. Copyright 2026 by the author(s).

## 1. Introduction

Large Language Models (LLMs) are increasingly deployed as *collaborative* Multi-Agent Systems (MAS), where multiple specialized agents communicate through a workflow to solve complex tasks (Wu et al., 2024a; Hong et al., 2023; Qian et al., 2024; Chen et al., 2023b; Tang et al., 2023). By decomposing problems such as code generation and mathematical reasoning into coordinated multi-agent interactions, MAS can outperform monolithic models. Importantly, MAS performance is shaped not only by the underlying LLMs but also by *system design* choices—including the workflow topology and, critically, the prompts that govern the behavior of each agent (Khattab et al., 2023).

Although some recent work explores automating workflow topology and agent-role design (Zhang et al., 2024; Zhuge et al., 2024; Hu et al., 2025; Liu et al., 2023; Yu et al., 2024), many real-world deployments rely on workflows that have undergone expert vetting, safety validation, and compliance review (e.g., medical SOPs and financial auditing) (Wu et al., 2025; Bodnari & Travis, 2025; Wells et al., 2025; Yang et al., 2024). Such workflows are therefore *rarely modified in practice*: even minor changes can trigger expensive re-validation procedures or may be prohibited altogether. Consequently, optimizing *agent-specific* prompts becomes the primary lever for improving system performance (Brown et al., 2020; Kojima et al., 2022).

However, prompt optimization for MAS presents a challenging combinatorial black-box optimization problem, characterized by three difficulties. **(i) Expensive evaluations:** evaluating a candidate prompt configuration requires end-to-end MAS execution, often involving multiple LLM calls, which severely limits the evaluation budget. **(ii) Topology-induced coupling:** changing an upstream prompt shifts the input distribution of downstream agents, making the objective non-separable and rendering independent optimization unstable. **(iii) Combinatorial search:** the joint prompt space is a discrete Cartesian product whose size grows exponentially with the number of agents, making exhaustive search infeasible. These constraints raise a question: *How can we perform sample-efficient and topology-aware prompt optimization for MAS under a combinatorial search space?*

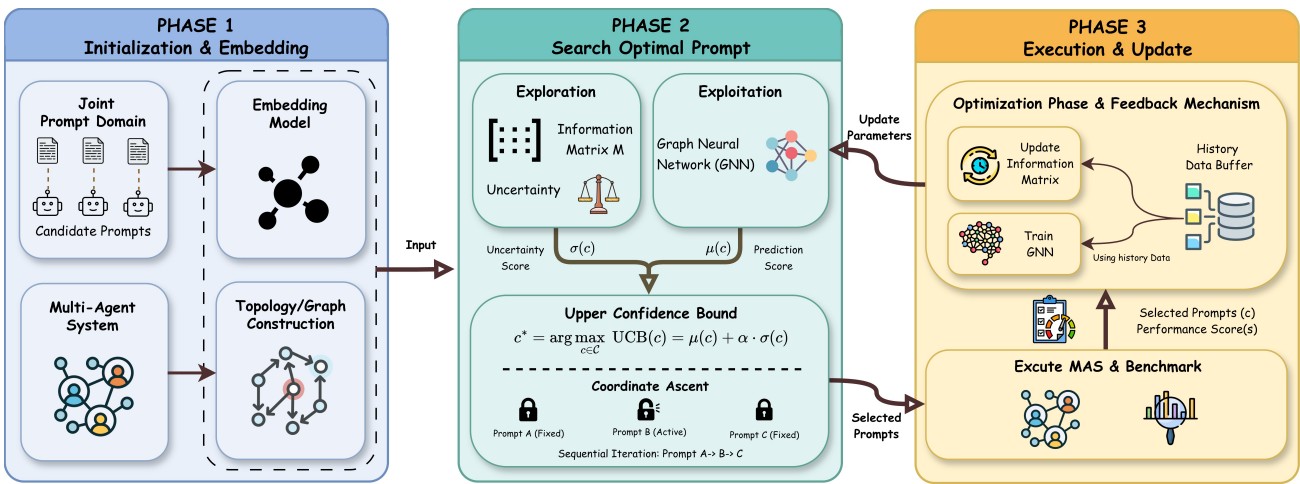

*Figure 1.* The MASPOB framework. (1) Initialization: Constructs agent topology and generates prompt embeddings. (2) Search: Selects optimal prompts via Coordinate Ascent, balancing exploitation (GNN prediction) and exploration (Linear UCB uncertainty). (3) Update: Refines the GNN model and information matrix using execution feedback.

Despite this motivation, existing prompt optimizers do not fully address the above setting. Single-agent optimizers (Guo et al., 2023; Lin et al., 2023; Chen et al., 2023a), such as OPRO (Yang et al., 2023) and PromptBreeder (Fernando et al., 2023), refine prompts in isolation and therefore ignore topology-induced coupling and the resulting distribution shift across agents. Multi-stage prompt optimizers like MIPRO (Opsahl-Ong et al., 2024b) apply Bayesian optimization to chained workflows, but typically capture dependencies only implicitly (e.g., via TPE) and remain largely topology-agnostic. Under costly evaluations and a combinatorial search space, this can be sample-inefficient and may miss *coordinated* prompt combinations that respect the MAS structure.

To address this gap, we propose **MASPOB** (**M**ulti-**A**gent **S**ystem **P**rompt **O**ptimization via **B**andits), which integrates uncertainty-guided exploration, topology-aware modeling, and scalable combinatorial search.

**Uncertainty-driven exploration.** To achieve sample efficiency under a strict evaluation budget, MASPOB frames prompt optimization as a contextual bandit problem (Kong et al., 2025; Lin et al., 2024; Wu et al., 2024b) and performs exploration using an Upper Confidence Bound (UCB) criterion. Specifically, it estimate epistemic uncertainty via an information matrix (Abbasi-Yadkori et al., 2011; Li et al., 2010; Chu et al., 2011) in the learned representation space, and combine this uncertainty with the surrogate's predicted performance to construct an acquisition score. This UCB-style objective prioritizes prompt configurations that are not only promising but also informative, enabling efficient allocation of limited evaluations.

**Topology-aware surrogate.** MASPOB employs a Graph

Neural Network (GNN) surrogate to explicitly encode the MAS workflow and model how prompt changes propagate along inter-agent edges. By representing agents as nodes and aggregating information via message passing, the surrogate provides a structural inductive bias beyond treating the MAS as an unstructured black box. The surrogate's topology-aware representations serve as features for the bandit objective, enabling more reliable generalization from limited evaluations (Zhang et al., 2025a).

**Scalable combinatorial search.** Finally, to mitigate combinatorial explosion, MASPOB adopts a Coordinate Ascent strategy that decomposes the global search into a sequence of tractable univariate updates while still accounting for inter-agent coupling through the topology-aware surrogate and the UCB objective.

In summary, our key contributions are as follows:

- We formalize prompt optimization for MAS as a budgeted black-box problem with topology-induced coupling and a discrete combinatorial search space, and identify key limitations of existing prompt optimizers in this setting.

- We propose MASPOB, combining a topology-aware GNN surrogate with uncertainty-guided bandit exploration and coordinate ascent to enable sample-efficient optimization under a strict evaluation budget.

- We evaluate MASPOB on six benchmarks, spanning question answering, code generation, and mathematical reasoning, and observe consistent improvements over strong baselines under matched evaluation budgets.

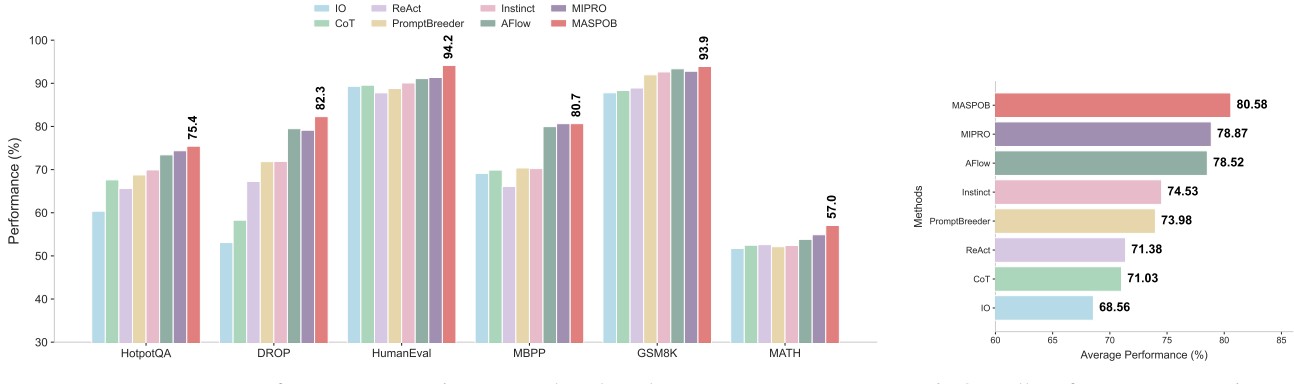

*(a)* Performance comparison across benchmarks.  *(b)* Overall performance comparison.

*Figure 2.* Performance evaluation of prompt optimization methods. (a) Detailed comparison across six diverse benchmarks including question answering (HotpotQA, DROP), code generation (HumanEval, MBPP), and mathematical reasoning (GSM8K, MATH). (b) Overall ranking based on average performance. MASPOB demonstrates superior performance with an average improvement of 12.02% over the IO baseline.

## 2. Problem Setting

We study prompt optimization for Large Language Model (LLM)-based Multi-Agent Systems (MAS), where multiple specialized agents collaborate to solve complex tasks. Each agent is powered by an LLM and is steered by a role-specific prompt; consequently, system-level performance emerges from inter-agent interactions rather than from any single prompt in isolation.

**MAS Workflow as a Directed Acyclic Graph.** We model an MAS workflow as a directed acyclic graph (DAG) $\mathcal{G} = (\mathcal{V}, \mathcal{E})$, where nodes $\mathcal{V} = \{a_1, a_2, \ldots, a_N\}$ denote $N$ agents and directed edges $\mathcal{E} \subseteq \mathcal{V} \times \mathcal{V}$ represent information flow. An edge $(a_i, a_j) \in \mathcal{E}$ means that the output produced by agent $a_i$ is included in the input context of agent $a_j$. For an input query $x$, agents execute in a topological order: each agent $a_i$ aggregates messages from its predecessors $\{a_j : (a_j, a_i) \in \mathcal{E}\}$, performs an LLM invocation, and forwards its output to downstream agents. The workflow terminates with a final system output $\hat{y}$. This DAG assumption applies only to inter-agent information flow within a single execution: individual agents may still perform retries, reflection, self-correction, or iterative refinement internally, while cross-agent feedback cycles are excluded.

**Prompt-Controlled Agent Behavior.** Each agent $a_i$ is associated with a prompt $p_i$ that acts as an instruction template specifying how the agent should interpret its inputs and format its outputs. Changing $p_i$ can alter the agent's reasoning strategy, output structure, and thus the quality of the information propagated to downstream agents. We denote by $\mathcal{P}_i$ the prompt domain (candidate set) for agent $a_i$. Because agents typically play different roles (e.g., reasoning, code generation, verification), their prompt domains can be heterogeneous, which further complicates prompt optimization

and motivates topology-aware modeling that accounts for inter-agent dependencies.

**Prompt Combination and Evaluation.** A *prompt combination* is a tuple $c = (p_1, p_2, \ldots, p_N)$ with $p_i \in \mathcal{P}_i$. The global search space is the Cartesian product $\mathcal{P} = \mathcal{P}_1 \times \mathcal{P}_2 \times \cdots \times \mathcal{P}_N$, whose size is $\prod_{i=1}^{N} |\mathcal{P}_i|$. Given a dataset $\mathcal{D} = \{(x, y)\}$ and an evaluation metric $m(\cdot, \cdot)$, we define the expected performance of the MAS under $c$ as

$$s(c \mid \mathcal{D}) = \mathbb{E}_{(x,y) \in \mathcal{D}}\big[m(\mathrm{MAS}(c, x), y)\big], \qquad (1)$$

where $\mathrm{MAS}(c, x)$ denotes the output produced by executing the MAS with prompt combination $c$ on input $x$.

**Optimization Objective.** Given a validation set $\mathcal{D}_v$ and a budget of $T$ evaluations, we seek the prompt combination that maximizes validation performance:

$$c^* = \arg\max_{c \in \mathcal{P}} s(c \mid \mathcal{D}_v). \qquad (2)$$

We then report the final test performance $s(c^* \mid \mathcal{D}_t)$ on a held-out test set $\mathcal{D}_t$. Overall, this is a combinatorial black-box optimization problem: $s(\cdot)$ can only be queried through expensive end-to-end MAS executions, while the search space grows exponentially with the number of agents.

**Budgeted Optimization Protocol.** In practice, evaluating $s(c \mid \mathcal{D}_v)$ is costly because it requires running the full end-to-end MAS workflow on validation instances. We assume a fixed budget $T$ and frame prompt optimization as a sequential, feedback-driven process over combinations $c^{(1)}, \ldots, c^{(T)}$, where validation scores guide subsequent selections. The key challenge is budget allocation: early evaluations explore diverse regions, while later evaluations exploit promising regions to refine the best-performing combination. This setting motivates modeling shared structure and inter-agent dependencies, rather than treating each combination independently.

# 3. Methodology

We present MASPOB, a framework for optimizing prompt combinations in Multi-Agent Systems. For a prompt combination $c = (p_1, \ldots, p_N)$, we encode each prompt $p_i$ into a $d$-dimensional embedding $\Phi(p_i) \in \mathbb{R}^d$ using a pre-trained text encoder, and represent the MAS workflow topology with an adjacency matrix $\mathbf{A}_{\mathcal{G}} \in \{0,1\}^{N \times N}$ derived from the dependency graph $\mathcal{G} = (\mathcal{V}, \mathcal{E})$.

Our framework consists of three components. (i) A topology-aware **Graph Neural Network (GNN)** surrogate predicts the performance of a prompt combination by modeling inter-agent dependencies; we instantiate it with a Graph Attention Network (GAT) (Veličković et al., 2017) that takes prompt embeddings as node features and the workflow topology as the graph structure (§3.1). (ii) We cast prompt search as a **contextual bandit** problem and apply LinUCB (Abbasi-Yadkori et al., 2011) to balance exploitation of high-scoring combinations with exploration of uncertain regions (§3.2). (iii) We use **coordinate ascent** to approximately maximize the UCB acquisition function by decomposing the joint search into a sequence of single-agent subproblems, reducing the per-iteration complexity from $O\left(\prod_{i=1}^{N} |\mathcal{P}_i|\right)$ to $O\left(\sum_{i=1}^{N} |\mathcal{P}_i|\right)$, where $N = |\mathcal{V}|$ is the number of agents (§3.3). Algorithm 1 summarizes the full procedure. For notational convenience, let $c_{-i}$ denote the prompts of all agents except agent $i$, and write $(c_{-i}, p)$ for the prompt combination obtained by setting the $i$-th prompt to $p \in \mathcal{P}_i$ while keeping the other prompts fixed.

## 3.1. Topology-Aware Performance Prediction

**Graph Construction.** We represent the MAS workflow as a directed graph where each node corresponds to an agent. For agent $a_i$, its node feature is initialized with the prompt embedding $\Phi(p_i) \in \mathbb{R}^d$. The adjacency matrix $\mathbf{A}_{\mathcal{G}}$ encodes the workflow dependencies, augmented with self-loops to facilitate information propagation across the entire workflow structure.

**Attention-Based Message Passing.** We instantiate the surrogate with a standard multi-head Graph Attention Network (GAT). Each layer uses $K$ attention heads to update node representations through attention-weighted message passing, enabling the model to learn the relative importance of different agent interactions.

For intermediate GAT layers, the outputs of each head are concatenated:

$$\mathbf{h}_i^{(l+1)} = \Big\|_{k=1}^{K} \sigma\left(\sum_{j \in \mathcal{N}(i) \cup \{i\}} \alpha_{ij}^{(k)} \mathbf{W}^{(l,k)} \mathbf{h}_j^{(l)}\right), \quad (3)$$

where $\mathcal{N}(i)$ denotes the neighbors of node $i$, $\mathbf{W}^{(l,k)}$ is the learnable projection matrix for attention head $k$ at layer $l$, $\sigma$

is a nonlinear activation function, and $\|$ denotes concatenation across heads.

For the final GAT layer, the outputs of all heads are averaged instead of concatenated:

$$\mathbf{h}_i^{(L)} = \sigma\left(\frac{1}{K} \sum_{k=1}^{K} \sum_{j \in \mathcal{N}(i) \cup \{i\}} \alpha_{ij}^{(k)} \mathbf{W}^{(L,k)} \mathbf{h}_j^{(L-1)}\right). \quad (4)$$

The attention coefficient $\alpha_{ij}^{(k)}$ for head $k$ quantifies the importance of agent $j$'s information to agent $i$. Following the standard GAT formulation, the unnormalized attention score is computed by applying a shared attention vector to the transformed target and source features:

$$e_{ij}^{(k)} = \text{LeakyReLU}\left(\mathbf{a}^{(k)\top}\left[\mathbf{W}^{(l,k)} \mathbf{h}_i^{(l)} \| \mathbf{W}^{(l,k)} \mathbf{h}_j^{(l)}\right]\right), \tag{5}$$

$$\alpha_{ij}^{(k)} = \frac{\exp\left(e_{ij}^{(k)}\right)}{\sum\limits_{r \in \mathcal{N}(i) \cup \{i\}} \exp\left(e_{ir}^{(k)}\right)}, \tag{6}$$

where $\mathbf{a}^{(k)}$ is a learnable attention vector for head $k$ and $\|$ denotes concatenation.

**Performance Prediction.** After $L$ layers of message passing, we obtain a graph-level representation through mean pooling over all node features:

$$\mathbf{z} = \frac{1}{N} \sum_{i=1}^{N} \mathbf{h}_i^{(L)}. \tag{7}$$

The predicted performance score is computed by a multilayer perceptron:

$$\mu(c) = \text{MLP}(\mathbf{z}) \in [0, 1]. \tag{8}$$

This prediction serves as the exploitation signal in our optimization framework, guiding the search toward promising regions of the prompt space.

**GNN Initialization.** MASPOB initializes the GNN surrogate through a limited warm-up phase: it randomly samples $T_0$ complete prompt combinations, evaluates them on the validation set, and trains the GAT predictor on the resulting observations $(\Phi(c), s(c \mid \mathcal{D}_v))$. This gives the surrogate an initial topology-aware performance estimate, while the subsequent UCB-guided search continues to explore high-uncertainty regions.

## 3.2. Bandit-Based Exploration-Exploitation Tradeoff

A fundamental challenge in prompt optimization is balancing exploitation of known good combinations with exploration of uncertain regions under a limited evaluation budget. We formulate this as a *contextual bandit* problem and adopt

Linear Upper Confidence Bound (LinUCB) to achieve a principled tradeoff. More details about the UCB design and additional empirical validation are provided in Appendix B.

**Uncertainty Quantification.** We maintain an information matrix $\mathbf{M} \in \mathbb{R}^{Nd \times Nd}$ that accumulates information from evaluated prompt combinations. Given a prompt combination $c$, we construct its combined embedding by concatenating individual prompt embeddings:

$$\Phi(c) = [\Phi(p_1); \Phi(p_2); \ldots; \Phi(p_N)] \in \mathbb{R}^{Nd}. \quad (9)$$

The information matrix is initialized as $\mathbf{M} = \lambda \mathbf{I}$ with regularization coefficient $\lambda$, and updated after each evaluation:

$$\mathbf{M} \leftarrow \mathbf{M} + \Phi(c)\Phi(c)^\top. \quad (10)$$

The uncertainty of a prompt combination is then estimated as:

$$\sigma(c) = \sqrt{\Phi(c)^\top \mathbf{M}^{-1} \Phi(c)}, \quad (11)$$

which measures the novelty of $c$ relative to previously evaluated combinations. Combinations in unexplored regions of the embedding space yield higher uncertainty values.

**Upper Confidence Bound.** The UCB acquisition function combines the predicted performance with an uncertainty bonus:

$$\mathrm{UCB}(c) = \mu(c) + \alpha \cdot \sigma(c), \quad (12)$$

where $\alpha > 0$ is the exploration coefficient. The first term encourages exploitation by favoring combinations with high predicted performance, while the second term promotes exploration by assigning higher scores to uncertain combinations. This principled tradeoff ensures efficient utilization of the limited evaluation budget.

### 3.3. Coordinate Ascent Search

Exhaustively evaluating all prompt combinations is computationally prohibitive, as the search space grows exponentially with the number of agents. To efficiently navigate this combinatorial space, we adopt coordinate ascent, which decomposes the joint optimization into a sequence of tractable single-agent optimizations.

Starting from the current best combination $c^* = (p_1^*, \ldots, p_N^*)$, we sequentially optimize the prompt for each agent while keeping others fixed:

$$p_i^* \leftarrow \arg\max_{p \in \mathcal{P}_i} \mathrm{UCB}(p_1^*, \ldots, p_{i-1}^*, p, p_{i+1}^*, \ldots, p_N^*). \quad (13)$$

This procedure reduces the per-iteration search complexity from $O(\prod_{i=1}^N |\mathcal{P}_i|)$ to $O(\sum_{i=1}^N |\mathcal{P}_i|)$, requiring UCB evaluations that scale linearly with the number of agents. Since UCB evaluation only requires forward passes through the

GAT model without actual MAS execution, this search process incurs negligible computational overhead compared to evaluating prompt combinations on the validation set.

For strongly coupled topologies, MASPOB can optionally use small block-coordinate updates over tightly connected agents. Rather than optimizing the full prompt tuple jointly, this variant updates only a local block, such as an upstream agent and a downstream agent connected by a workflow edge, while keeping the remaining prompts fixed. This preserves the efficiency of coordinate search while allowing the acquisition function to capture stronger prompt interactions along direct dependencies. We use single-agent updates by default, and provide block-coordinate details and empirical analysis in Appendix C.

---

**Algorithm 1** MASPOB: Multi-Agent System Prompt Optimization with Bandit

---

**Require:** MAS workflow $\mathcal{G} = (\mathcal{V}, \mathcal{E})$; prompt domains $\{\mathcal{P}_i\}_{i=1}^N$; validation set $\mathcal{D}_v$; warm-up rounds $T_0$; total budget $T$; exploration coefficient $\alpha$; regularization $\lambda$

**Ensure:** Optimized prompt combination $c^*$

1: Compute prompt embeddings $\Phi(p)$ for all $p \in \bigcup_{i=1}^N \mathcal{P}_i$
2: Construct adjacency matrix $\mathbf{A}_\mathcal{G}$ from workflow $\mathcal{G}$
3: Initialize information matrix $\mathbf{M} \leftarrow \lambda \mathbf{I}$; history $\mathcal{H} \leftarrow \emptyset$
4: Initialize best pair $(s^*, c^*) \leftarrow (0, \text{null})$
5: **for** $t = 1$ to $T_0$ **do**
6:     Randomly sample a full combination $c$ and obtain score $s(c \mid \mathcal{D}_v)$ via end-to-end MAS execution
7:     Update $\mathcal{H}$, $\mathbf{M}$, and incumbent $(s^*, c^*)$
8: **end for**
9: Train the GAT on the warm-up observations in $\mathcal{H}$
10: **for** $t = T_0 + 1$ to $T$ **do**
11:     *// Coordinate ascent with UCB guidance*
12:     Initialize search point $c \leftarrow c^*$ for coordinate ascent
13:     **for** $i = 1$ to $N$ **do**
14:         *// GNN prediction (exploitation)*
15:         $\mu_p \leftarrow \mathrm{GAT}(\Phi(c_{-i}, p), \mathbf{A}_\mathcal{G})$ for all $p \in \mathcal{P}_i$
16:         *// UCB uncertainty (exploration)*
17:         $\sigma_p \leftarrow \sqrt{\Phi(c_{-i}, p)^\top \mathbf{M}^{-1} \Phi(c_{-i}, p)}$
18:         $p_i \leftarrow \arg\max_{p \in \mathcal{P}_i} [\mu_p + \alpha \cdot \sigma_p]$
19:         Update $c \leftarrow (c_{-i}, p_i)$
20:     **end for**
21:     *// Evaluation and model update*
22:     Obtain $s(c \mid \mathcal{D}_v)$ by executing the MAS with prompt combination $c$
23:     $\mathcal{H} \leftarrow \mathcal{H} \cup \{(\Phi(c), s(c \mid \mathcal{D}_v))\}$
24:     $\mathbf{M} \leftarrow \mathbf{M} + \Phi(c)\Phi(c)^\top$ *// update information matrix*
25:     Retrain the GAT predictor on $\mathcal{H}$ *// update parameters*
26:     **if** $s(c \mid \mathcal{D}_v) > s^*$ **then**
27:         $s^* \leftarrow s(c \mid \mathcal{D}_v)$; $c^* \leftarrow c$
28:     **end if**
29: **end for**
30: **return** $c^*$

---

# 4. Experiments

## 4.1. Experiment Setup

**Datasets.** Following established practices in agentic workflows (Zhang et al., 2024), we evaluate on six widely used public benchmarks: HotpotQA (Yang et al., 2018), DROP (Dua et al., 2019), HumanEval (Chen et al., 2021), MBPP (Austin et al., 2021), GSM8K (Cobbe et al., 2021), and MATH (Hendrycks et al., 2021). These benchmarks span question answering, code generation, and mathematical reasoning. For fair comparisons, We follow AFlow's dataset construction protocol, but use a different validation/test split. Additional dataset details are provided in Appendix A.1.

**Baselines.** We compare MASPOB with representative methods from three categories: $(i)$ single-agent prompting without prompt optimization, including IO (a direct LLM call), CoT (Wei et al., 2022), and ReAct (Yao et al., 2022); $(ii)$ single-agent prompt optimization, including PromptBreeder (Fernando et al., 2023) and Instinct (Lin et al., 2023); and $(iii)$ multi-agent systems, including AFlow (Zhang et al., 2024) and MIPRO (Opsahl-Ong et al., 2024b) (a multi-agent *prompt optimization* baseline). More details can be seen in Appendix A.2

**Metrics.** We use standard task-specific metrics. For mathematical reasoning benchmarks (GSM8K and $MATH_{lv5*}$), we report the solve rate (%). For code generation (HumanEval and MBPP), we report Pass@1 following (Chen et al., 2021). For question answering (HotpotQA and DROP), we report the F1 score.

**Implementation Details.** We use Qwen3-Embedding-8B (Zhang et al., 2025b) as the text embedding model and GPT-4o-mini (Hurst et al., 2024) as the backbone LLM for agent execution, unless otherwise specified. For fairness, each prompt optimization method is given the same validation budget of 50 evaluations to select the best prompt combination (i.e., the one with the highest validation performance). We then evaluate the selected combination three times on the test set and report the mean as the final score. Additional implementation details are provided in Appendix A.3.

## 4.2. Experimental Results and Analysis

**Main Results.** Table 1 reports test performance across all benchmarks. MASPOB achieves the best result on every benchmark, with an average score of 80.58%. Compared with IO, AFlow, and MIPRO, MASPOB improves the average score by 12.02%, 2.06%, and 1.71%, respectively. Under the same validation budget of 50 evaluations per method, these results indicate that MASPOB is more sample-efficient and consistently outperforms the existing baselines in terms of overall performance. Beyond average gains, MASPOB improves both reasoning-heavy tasks (e.g., hotpotQA and

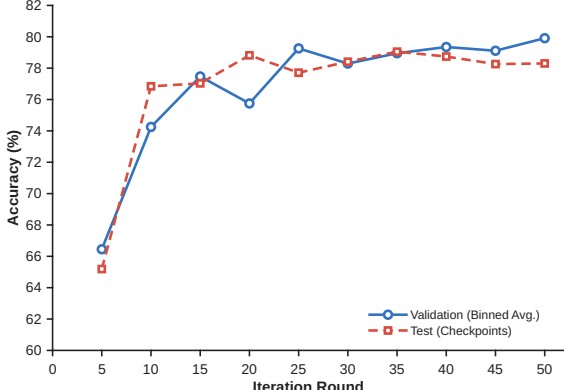

*Figure 3.* Optimization convergence on validation and test sets. The curves show the average validation accuracy, computed by averaging over every five rounds, and the test accuracy at rounds 5, 10, . . . , 50. For each selected test combination, the accuracy at these rounds is evaluated and averaged over three runs.

math) and structured-output tasks (e.g., code generation). This suggests that the benefit is not limited to a single task format, but instead comes from better coordination among agents.

**Convergence Trend.** Figure 3 illustrates convergence on both the validation set and the corresponding test set performance of the prompt combination selected at different rounds. MASPOB improves steadily as the number of evaluations increases, and the selected prompt combination stabilizes on the test set at round 35. This indicates that MASPOB can identify well-coordinated prompts early in the search process, achieving near-optimal performance with relatively few validation evaluations under a fixed budget.

**Generalization to Complex MAS Structures.** To assess robustness to more complex workflows, we evaluate all methods on MAS structures with higher topological complexity. As shown in Table 2, MASPOB remains the best-performing method, indicating strong generalization across different workflow complexities.

In this setting, MIPRO underperforms AFlow, likely because its TPE optimizer only implicitly captures inter-variable dependencies from past evaluations and does not explicitly leverage the DAG topology. In contrast, MASPOB explicitly encodes the workflow structure via the GNN surrogate, enabling more efficient identification of coordinated prompt combinations.

Finally, MASPOB performs better on the original (simpler) workflows than on the more complex counterparts across benchmarks. This suggests that, in many practical settings, a simple but well-designed workflow may suffice, and prompt optimization alone (without modifying the workflow structure) can yield substantial gains.

*Table 1.* Performance comparison across six benchmarks. We report mean accuracy (%) ± standard deviation over three runs. Bold indicates the best result for each benchmark.

| Method | QA | | Code | | Math | | |
| --- | --- | --- | --- | --- | --- | --- | --- |
| | HotpotQA | DROP | HumanEval | MBPP | GSM8K | MATH | Avg. |
| IO (GPT-4o-mini) | 60.36±0.48 | 53.09±0.36 | 89.31±2.02 | 69.11±1.19 | 87.80±0.44 | 51.71±0.43 | 68.56 |
| CoT (Wei et al., 2022) | 67.62±0.48 | 58.27±0.42 | 89.57±1.16 | 69.89±1.19 | 88.34±0.18 | 52.47±1.25 | 71.03 |
| ReAct (Yao et al., 2022) | 65.61±0.22 | 67.25±0.85 | 87.79±1.32 | 66.08±1.48 | 88.91±0.24 | 52.61±0.62 | 71.38 |
| PromptBreeder (Fernando et al., 2023) | 68.76±0.18 | 71.85±0.87 | 88.80±2.45 | 70.38±0.36 | 91.97±0.80 | 52.13±0.97 | 73.98 |
| Instinct (Lin et al., 2023) | 69.92±0.17 | 71.90±0.53 | 90.08±2.76 | 70.23±1.14 | 92.64±0.39 | 52.40±1.52 | 74.53 |
| AFlow (Zhang et al., 2024) | 73.42±0.38 | 79.48±0.12 | 91.09±0.44 | 79.96±0.67 | 93.36±0.43 | 53.83±0.40 | 78.52 |
| MIPRO (Opsahl-Ong et al., 2024a) | 74.37±1.07 | 79.13±0.59 | 91.35±0.44 | **80.65**±0.36 | 92.80±0.42 | 54.90±0.61 | 78.87 |
| **MASPOB** | **75.43**±0.27 | **82.28**±0.55 | **94.15**±0.44 | **80.65**±0.29 | **93.90**±0.15 | **57.05**±0.51 | **80.58** |

*Table 2.* Performance on more complex MAS structures. We generate structures with greater topological complexity using AFlow (8, 7, and 7 agents for HotpotQA, DROP, and HumanEval, respectively), compared to the original simpler topologies (3, 2, and 3 agents). All other experimental settings are kept unchanged. We report mean accuracy (%) ± standard deviation over three runs. **Bold** indicates the best result for each benchmark.

| Method | HotpotQA | DROP | HumanEval | Avg. |
| --- | --- | --- | --- | --- |
| Aflow | 73.38± 0.36 | 80.28± 1.07 | 89.57± 0.44 | 81.08 |
| MIPRO | 74.03± 0.38 | 79.99± 0.78 | 86.77± 1.16 | 80.26 |
| **MASPOB** | **74.43**± 0.37 | **81.55**± 1.29 | **90.08**± 0.76 | **82.02** |

**Generalization to Different Prompt Domains.** To assess robustness to prompt-domain construction, we evaluate MASPOB on an alternative candidate set produced by MIPRO, which generates prompt domain via different strategies (data-aware, program-aware, fewshot-aware, and tip-aware). As shown in Table 5, the results are close across the two domains, suggesting that MASPOB's gains mainly come from topology-aware contextual-bandit optimization rather than from a single prompt-domain construction recipe. At the same time, the quality and diversity of candidate prompts can still affect absolute performance, since the optimizer can only select from the provided prompt domain. We also present the construction details of our prompt domain in Appendix A.3.

## 5. Ablation Study

**Effectiveness of Graph Neural Networks.** To quantify the contribution of the GNN surrogate in modeling topology-induced interactions during prompt search, we replace the GNN with a multi-layer perceptron (MLP) while keeping all other components unchanged. As shown in Table 3, removing the GNN reduces the average performance by 2.31%, resulting in slightly worse performance than AFlow (by 0.2%). This indicates that explicitly modeling the workflow

topology is important for capturing inter-agent couplings and identifying coordinated prompt combinations.

Beyond overall averages, the GNN yields consistent gains across benchmarks, indicating that topology-aware inductive bias remains beneficial even as the number of agents and prompt domains vary.

The performance drop is larger on tasks with rich intermediate messages (e.g., multi-hop QA and multi-step math), where downstream agents heavily depend on complete and well-structured context; here, topology-aware message passing provides a stronger inductive bias than treating prompts as an unordered vector.

**Generalization to Other LLMs.** To test whether the gains of MASPOB transfer to different backbone LLMs, we replace GPT-4o-mini with Qwen3-32B (Yang et al., 2025) and keep all other settings unchanged. Table 4 shows that MASPOB remains the best-performing method across all three benchmarks, suggesting that the improvements are not specific to a single LLM. This further supports that MASPOB mainly improves *prompt coordination* across agents, rather than exploiting idiosyncrasies of a particular model.

**Coordinate Ascent vs. Global Search.** To evaluate the effectiveness of coordinate ascent for optimizing the acquisition function, we compare it with a global-search baseline that enumerates all prompt combinations at each iteration. As shown in Table 6 and Figure 4, coordinate ascent achieves performance comparable to global search, with only minor degradation, while reducing runtime by 99.8% on HotpotQA and 98% on DROP. This demonstrates a favorable trade-off between optimization quality and computational cost.

## 6. Related Work

**Prompt Optimization.** Prompt optimization has evolved from manual, heuristic prompt engineering to automated

*Table 3.* Ablation study replacing the GNN with an MLP. We replace the GNN module with a multi-layer perceptron (MLP) while keeping all other components unchanged. We report mean accuracy (%) ± standard deviation over three runs. **Bold** indicates the best result for each benchmark.

| Method | HotpotQA | DROP | HumanEval | MBPP | GSM8K | MATH | Avg. |
|---|---|---|---|---|---|---|---|
| AFlow | $73.42_{\pm0.38}$ | $79.48_{\pm0.12}$ | $91.09_{\pm0.44}$ | $79.96_{\pm0.67}$ | $93.36_{\pm0.43}$ | $53.83_{\pm0.40}$ | 78.52 |
| MIPRO | $74.37_{\pm1.07}$ | $79.13_{\pm0.59}$ | $91.35_{\pm0.44}$ | $\mathbf{80.65}_{\pm0.36}$ | $92.80_{\pm0.42}$ | $54.90_{\pm0.61}$ | 78.87 |
| **MASPOB** | $\mathbf{75.43}_{\pm0.27}$ | $\mathbf{82.28}_{\pm0.55}$ | $\mathbf{94.15}_{\pm0.44}$ | $\mathbf{80.65}_{\pm0.29}$ | $\mathbf{93.90}_{\pm0.15}$ | $\mathbf{57.05}_{\pm0.51}$ | $\mathbf{80.58}$ |
| w/o GNN | $74.38_{\pm0.44}$ | $79.20_{\pm0.64}$ | $90.33_{\pm1.08}$ | $79.18_{\pm0.77}$ | $92.95_{\pm0.31}$ | $53.57_{\pm0.83}$ | 78.27 |
| Δ (GNN Gain) | +1.05 | +3.08 | +3.82 | +1.47 | +0.95 | +3.48 | **+2.31** |

*Table 4.* Generalization to Qwen-3-32B. Results using Qwen-3-32B as the backbone LLM, replacing GPT-4o-mini while keeping all other settings unchanged. We report mean accuracy (%) ± standard deviation over three runs. **Bold** indicates the best result for each benchmark.

| Method | DROP | HumanEval | MATH | Avg. |
|---|---|---|---|---|
| Aflow | $84.73_{\pm0.48}$ | $92.11_{\pm0.44}$ | $69.97_{\pm0.32}$ | 82.27 |
| MIPRO | $85.28_{\pm0.02}$ | $93.64_{\pm0.54}$ | $70.70_{\pm1.25}$ | 83.21 |
| **Our** | $\mathbf{85.31}_{\pm0.47}$ | $\mathbf{94.15}_{\pm0.44}$ | $\mathbf{70.84}_{\pm0.48}$ | $\mathbf{83.43}$ |

*Table 5.* Prompt-domain robustness. We replace the MASPOB prompt domain with MIPRO's while keeping the MASPOB optimizer unchanged. We report mean accuracy (standard deviation over three runs. Bold indicates the best result for each benchmark.

| Prompt Domain | DROP | MATH | Avg. |
|---|---|---|---|
| MIPRO domain | $\mathbf{82.81}_{\pm0.15}$ | $\mathbf{57.45}_{\pm0.61}$ | $\mathbf{70.13}$ |
| MASPOB domain | $82.28_{\pm0.55}$ | $57.05_{\pm0.51}$ | 69.67 |

*Table 6.* Performance and runtime comparison between Coordinate Ascent and Global Search. We replace the coordinate ascent strategy with an exhaustive global search on the HotpotQA and DROP benchmarks, while keeping all other experimental settings unchanged. We report mean accuracy (%) ± standard deviation over three runs.

| Method | HotpotQA | | DROP | |
|---|---|---|---|---|
| | Perf.(%) | Time(s) | Perf.(%) | Time(s) |
| Coordinate | $75.43_{\pm0.27}$ | 15.9 | $82.28_{\pm0.55}$ | 7.7 |
| Global | $75.72_{\pm0.29}$ | 8801 | $82.76_{\pm0.27}$ | 392 |
| Δ | -0.29 | **99.8%↓** | -0.48 | **98.0%↓** |

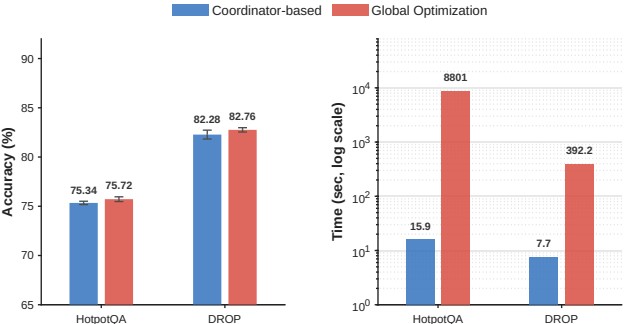

*Figure 4.* Performance and runtime comparison between coordinate ascent and global search. The figure illustrates optimization trajectories and time costs on selected benchmarks.

optimization methods. Prior work has shown that prompting strategies such as Chain-of-Thought (CoT) (Wei et al., 2022), Tree-of-Thoughts (ToT) (Yao et al., 2023a), ReAct (Yao et al., 2023b), and Self-Consistency (Wang et al., 2023b) can substantially improve the reasoning capability of large language models (LLMs). To reduce human effort, a growing body of work studies automated prompt optimization. Early methods such as APE (Zhou et al., 2022) and OPRO (Yang et al., 2023) use LLMs as optimizers. Later approaches explored evolutionary search (e.g., PromptBreeder (Fernando et al., 2023), EvoPrompt (Guo et al., 2023)), gradient-inspired or gradient-based methods (e.g., TextGrad (Yuksekgonul et al., 2024), ProTeGi (Pryzant et al., 2023)), as well as edit-based search (Prasad et al., 2023) and reinforcement learning (Deng et al., 2022) for discrete prompt tuning. Beyond text-only LLMs, related optimization objectives have also been explored in multimodal settings, where stabilizing vision–language representations can improve zero-shot adversarial robustness (Dong et al., 2025).

Most of these methods target single-agent settings. For multi-stage pipelines, MIPRO (Opsahl-Ong et al., 2024a), built on the DSPy framework (Khattab et al., 2023), performs multi-stage optimization of instructions and few-shot demonstrations via Bayesian optimization over modular programs. Since an MAS can be viewed as a computation graph of LLM calls, MIPRO is a strong and natural baseline. However, generic multi-stage optimizers are often challenged by topology-induced dependencies in MAS: small changes to upstream prompts can shift the input distribution of downstream agents, leading to cascading effects and a non-stationary optimization landscape.

**MAS Workflows.** Agentic frameworks such as Auto-Gen (Wu et al., 2024a), MetaGPT (Hong et al., 2023), CAMEL (Li et al., 2023), and ChatDev (Qian et al., 2024) have enabled the construction of complex collaboration workflows. Much recent work studies *structural search*, which aims to discover effective interaction graphs and team compositions dynamically (e.g., GPTSwarm (Zhuge et al., 2024), AFlow (Zhang et al., 2024), and DyLAN (Liu et al., 2023)).

In contrast, many high-stakes industrial applications (e.g., financial auditing, medical diagnosis, and automated standard operating procedures) require workflows to follow expert-validated topologies for compliance and auditability (Pei et al., 2025; Nandi et al., 2025; Ye et al., 2025). Such *frozen-topology* settings are not well served by structural search, as modifying the workflow structure may be infeasible or undesirable. Semantic Backpropagation (Wang et al., 2024) is also closely related, as it treats language-based agentic systems as computational graphs and propagates semantic feedback backward to optimize node-level prompts or instructions. The closest prior work to ours is MAPRO (Zhang et al., 2025c), which formulates multi-agent prompt optimization as a maximum a posteriori (MAP) inference problem and proposes a language-guided variant of belief propagation. While MAPRO explicitly accounts for inter-agent dependencies and provides a principled formulation, it relies on intricate inference procedures that may raise efficiency concerns in large combinatorial spaces.

Beyond inference complexity, a major bottleneck in optimizing such MAS is the prohibitive evaluation cost—carrying out a single evaluation of systems like Voyager (Wang et al., 2023a) requires numerous LLM API calls, rendering data-intensive techniques like RL or large-scale evolution (Guo et al., 2023) impractical. While sample-efficient methods like Contextual Bandits (e.g., LinUCB (Li et al., 2010)) offer a potential solution, existing algorithms are mostly "structure-blind" in that they treat the search space as independent vectors, ignoring the strong topological priors in MAS. Although GNNs have demonstrated strong structure modeling capabilities in combinatorial optimization (Li et al., 2018), a bandit method that effectively leverages GNNs to navigate the structured search space of fixed-topology MAS remains a missing piece.

## 7. Conclusion and Future Work

We study prompt optimization for LLM-based multi-agent systems (MAS), where multiple LLM-driven agents interact over a fixed workflow topology and must be evaluated end-to-end at high cost. We propose **MASPOB**, a sample-efficient optimizer that models topology-induced prompt coupling with a GNN surrogate and guides search via a LinUCB-style bandit and coordinate ascent. Across six benchmarks spanning QA, code generation, and mathematical reasoning, MASPOB consistently outperforms strong single-agent and multi-agent baselines under the same evaluation budget. Ablations further confirm that explicit topology encoding and uncertainty-aware exploration are key contributors to the observed gains, suggesting that prompt optimization alone can deliver meaningful improvements without altering expert-designed workflows.

**Outlook.** Beyond improving average accuracy, our results highlight the importance of exploiting workflow structure when optimizing multi-agent prompts under tight budgets. We believe MASPOB provides a practical foundation for deploying topology-aware prompt optimization in real-world agentic systems, where workflows are often specified by domain experts and must remain auditable and reproducible.

**Limitations.** MASPOB assumes a static DAG for inter-agent information flow, which matches our target setting of frozen-topology MAS deployment where workflows are designed by experts and kept auditable. This assumption still allows intra-agent retry, reflection, and refinement loops, but does not explicitly model cross-agent feedback cycles or dynamic conditional paths within a single execution. Extending MASPOB to such cyclic or dynamically routed inter-agent workflows is an important direction for future work, for example by estimating conditional or recurrent edge weights and incorporating them into condition-aware message passing.

## Impact Statement

This paper presents work whose goal is to advance the field of Machine Learning. There are many potential societal consequences of our work, none which we feel must be specifically highlighted here.

## Acknowledgements

This work was supported in part by the National Natural Science Foundation of China (Grant No. 62506319), the Guangdong Basic and Applied Basic Research Foundation (Grant No. 2026A1515030032), the Shenzhen Science and Technology Program (Grant No. JCYJ20250604141031003) and the Pearl River Talent Program of Guangdong Province (Grant No. 2024QN11X069).

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

# A. Experimental Details

## A.1. Dataset Partitioning

We use six public benchmarks in our experiments. For each benchmark, we construct validation and test splits with a validation-to-test ratio between 1:4 and 1:8. We use the full datasets for GSM8K, HumanEval, and MBPP. For HotpotQA and DROP, we randomly sample 1,000 instances from each dataset, following (Hu et al., 2025). For MATH, we follow (Hong et al., 2023) and select 617 level-5 problems from four representative categories (Combinatorics & Probability, Number Theory, Pre-algebra, and Pre-calculus). Table 7 summarizes the resulting split sizes.

| Dataset | Validation | Test |
|---|---|---|
| DROP | 100 | 800 |
| HotpotQA | 100 | 800 |
| GSM8K | 150 | 1,055 |
| MATH | 100 | 486 |
| MBPP | 86 | 341 |
| HumanEval | 33 | 131 |

*Table 7.* Dataset splits used in our experiments. Validation samples are used for prompt optimization, while test samples are held out for final evaluation.

## A.2. Baselines

We compare MASPOB against both single-agent and multi-agent baselines.

**Single-Agent Baselines.** For single-agent baselines without prompt optimization, **IO** (input–output) directly queries the LLM in a single pass, without additional reasoning structures. Optimized single-agent baselines include **Prompt-Breeder** (Fernando et al., 2023) and **Instinct** (Lin et al., 2023), which evolve or refine prompts using the LLM itself via evolutionary or gradient-free optimization strategies.

**Multi-Agent Baselines.** For multi-agent baselines, we first generate MAS structures (including the number of agents, roles, initial prompts, and topology) using **AFlow** (Zhang et al., 2024) with default settings. We then keep the structure fixed, and run **MASPOB** (ours) and **MIPRO** (Opsahl-Ong et al., 2024a) to optimize prompts on top of this structure. This setup isolates the effect of prompt optimization from workflow design.

## A.3. Implementation Details

We provide the hyperparameter settings used in our experiments for reproducibility. Unless otherwise specified, all experiments use GPT-4o-mini for both prompt generation and workflow execution. Table 8 summarizes the hyperparameters used in MASPOB.

**Prompt Domain Generation** For each agent, we construct a prompt domain $\mathcal{P}_i$ containing 20 prompt variants. The first prompt is the original template generated from AFlow, serving as the baseline. The remaining 19 variants are generated via LLM-based paraphrasing using GPT-4o-mini. Rather than treating prompt generation as unconstrained free-form rewriting, we construct role-aligned alternatives that preserve each agent's semantic intent, required input/output placeholders, and workflow-specific responsibilities. This design is important for MAS settings because agents typically have predefined roles such as planner, solver, verifier, programmer, or formatter; changing the role itself would confound prompt optimization with workflow redesign.

To increase diversity within this role-preserving space, we use a style-controlled rewriting process. Specifically, we define multiple style dimensions, such as reasoning style, output format, verification strategy, error-handling behavior, instruction explicitness, and conciseness. We then sample style configurations and convert them into rewriting instructions for GPT-4o-mini, producing candidate prompts that vary along task-relevant directions while maintaining the original agent role and all required placeholders. This yields a diverse but well-scoped candidate set for each agent. The prompt-domain robustness experiment in Section 4 further evaluates MASPOB under an alternative candidate set produced by MIPRO.

| Category | Parameter | Value |
|---|---|---|
| Optimization | Maximum optimization rounds | 45 |
| | Prompt variants per agent | 20 |
| | Pretraining rounds | 5 |
| | Search strategy | Coordinate |
| | Random seed | 42 |
| UCB | Exploration coefficient $\alpha$ | 0.2 |
| | Regularization coefficient $\lambda$ | 1.0 |
| | Fisher matrix update coefficient | 10 |
| | UCB type | Linear |
| GNN Architecture | Hidden dimension | 32 |
| | Number of GAT layers | 1 |
| | Dropout rate | 0.05 |
| | Learning rate | $5 \times 10^{-3}$ |
| | Weight decay | $1 \times 10^{-5}$ |
| | Pretraining epochs | 800 |
| | Early stopping patience | 200 |
| Test Evaluation | Test repetitions | 3 |
| | Concurrent API calls | 50 |

*Table 8.* Hyperparameter configurations for MASPOB.

*Table 9.* Ablation on uncertainty estimation. We replace MASPOB's LinUCB-style linear uncertainty bonus with a neural uncertainty estimator while keeping all other components fixed. Results are mean accuracy (%) $\pm$ standard deviation over three runs; best results are bolded.

| Method | HotpotQA | DROP | HumanEval | Avg. |
|---|---|---|---|---|
| AFlow | 73.42±0.31 | 79.48±0.10 | 91.09±0.36 | 81.33 |
| MIPRO | 75.09±0.25 | 80.91±0.37 | 91.35±0.36 | 82.45 |
| **MASPOB** | **75.43±0.17** | **82.28±0.45** | **94.15±0.36** | **83.95** |
| w/ Neural U. | 74.42±0.15 | 78.84±0.47 | 91.35±0.36 | 81.54 |
| $\Delta$ | -1.01 | -3.44 | -2.80 | **-2.41** |

*Table 10.* Sensitivity to the warm-up budget. We vary the number of initial evaluations used to initialize the GNN surrogate while keeping the total budget fixed at 50 evaluations and leaving all other optimizer settings unchanged. This study tests whether MASPOB needs many random initial observations before UCB-guided search begins. Results are mean accuracy (%) $\pm$ standard deviation over three runs; best results are bolded.

| Warm-up | DROP | MATH | Avg. |
|---|---|---|---|
| 1 | 79.34±0.15 | 56.35±0.77 | 67.85 |
| 5 | **82.28±0.55** | **57.05±0.51** | **69.67** |
| 10 | 82.60±0.31 | 54.67±0.25 | 68.64 |

**Prompt Embeddings.** We use Qwen3-Embedding-8B to obtain 1024-dimensional embeddings for each prompt variant. These embeddings are used as the feature representation in the bandit model and as node features for the GNN surrogate.

## B. Why Linear (vs. Neural) Uncertainty?

MASPOB follows a NeuralLinear-style decomposition: the GNN surrogate provides an expressive topology-aware representation for exploitation, while the LinUCB bonus estimates uncertainty in this learned representation space using a closed-form information matrix. This design avoids relying on the GNN itself to produce calibrated uncertainty from very few observations. Neural uncertainty estimators (Zhou et al., 2020; Lin et al., 2024; Wu et al., 2024b; Lin et al., 2023) (e.g., NeuralUCB-style methods that form a UCB bonus using neural features, often instantiated as the gradient of the network output with respect to its parameters under a local linearization/NTK interpretation) can be more expressive than linear models, but they are less suitable for our low-budget setting. Each bandit round requires an expensive end-to-end MAS execution, and with only 50 evaluations, neural uncertainty estimates can be poorly calibrated, sensitive to training instability, and slow to converge. In contrast, the LinUCB bonus provides a stable and computationally lightweight uncertainty estimate via the information matrix, and integrates naturally with coordinate ascent, which requires many inexpensive acquisition evaluations per round. Empirically, replacing the LinUCB-style linear uncertainty with a neural uncertainty estimator consistently reduces performance (Table 9) and leads to much slower uncertainty reduction over 45 rounds (Figure 5).

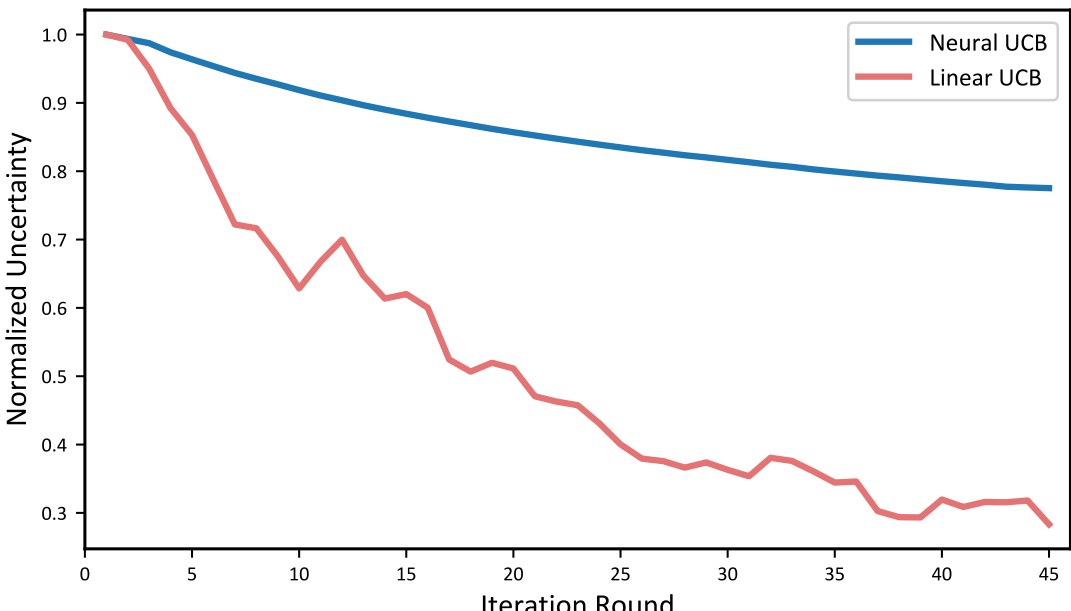

*Figure 5.* Uncertainty convergence over optimization rounds. Linear uncertainty decreases by 71.68%, whereas neural uncertainty decreases by 22.48% within 45 rounds, suggesting that neural uncertainty may require more exploration to reach comparable confidence.

## C. Additional Robustness and Sensitivity Analyses

This section provides additional robustness and sensitivity analyses for MASPOB. We examine five implementation choices: the warm-up budget, prompt embedding backbone, uncertainty approximation, coordinate-search granularity, and UCB exploration coefficient. Unless otherwise specified, all experiments follow the main protocol: we keep the validation budget, backbone LLM, workflow structure, and evaluation procedure fixed, and vary only the factor under study. This controlled design isolates the effect of each component and tests whether MASPOB's gains remain stable under practical design variations.

### C.1. Warm-up Sensitivity

We vary the number of warm-up rounds while keeping the total budget fixed at 50 evaluations. As shown in Table 10, five warm-up rounds achieve the best average performance, whereas using either fewer or more warm-up evaluations lowers the final score. This suggests that a moderate warm-up budget is sufficient to initialize the GNN surrogate, while leaving enough evaluations for UCB-guided optimization. Notably, MASPOB remains competitive even with only one warm-up round, indicating that the UCB bonus can partially compensate for early surrogate uncertainty.

### C.2. Embedding-Model Sensitivity

We evaluate MASPOB with three prompt embedding backbones: MPNet, BGE-Large, and Qwen3-8B. Table 11 shows that stronger embeddings generally lead to better optimization performance, with Qwen3-8B achieving the best average score. At the same time, MPNet and BGE-Large remain competitive and still exceed the corresponding AFlow and MIPRO averages of 66.66 and 67.02 under the same DROP–MATH setting. These results suggest that MASPOB benefits from high-quality semantic representations, but does not rely on a single embedding model to outperform strong baselines.

### C.3. Diagonal LinUCB for Larger-Scale Settings

For larger MAS workflows, inverting the full LinUCB information matrix can become computationally expensive. We therefore evaluate a diagonal approximation that reduces the cost of uncertainty estimation. Table 12 shows that this approximation underperforms full Linear UCB, as expected, because it ignores feature correlations. Nevertheless, its average score of 67.90 still exceeds the corresponding AFlow and MIPRO averages of 66.66 and 67.02 under the same

*Table 11.* Sensitivity to the prompt-embedding backbone. We compare three embedding models for constructing prompt features while keeping the optimizer, workflow, and evaluation budget unchanged. Results are mean accuracy (%) $\pm$ standard deviation over three runs; best results are bolded.

| Embedding | DROP | MATH | Avg. |
|---|---|---|---|
| MPNet | 80.46$\pm$0.22 | 56.76$\pm$0.74 | 68.61 |
| BGE-Large | 82.20$\pm$0.22 | 55.14$\pm$0.44 | 68.67 |
| Qwen3-8B | **82.28$\pm$0.55** | **57.05$\pm$0.51** | **69.67** |

*Table 12.* Full versus diagonal Linear UCB. The diagonal variant approximates the uncertainty matrix to reduce estimation cost while keeping the remaining MASPOB components unchanged. This comparison isolates the accuracy–efficiency tradeoff of the uncertainty approximation. Results are mean accuracy (%) $\pm$ standard deviation over three runs; best results are bolded.

| UCB Type | DROP | MATH | Avg. |
|---|---|---|---|
| Diag-Linear | 80.04$\pm$0.46 | 55.76$\pm$0.62 | 67.90 |
| Linear | **82.28$\pm$0.55** | **57.05$\pm$0.51** | **69.67** |

*Table 13.* Block-coordinate search under stronger coupling. Block size 1 is standard coordinate ascent, whereas block size 2 jointly updates connected agents while keeping the search budget fixed. Results are mean accuracy (%) $\pm$ standard deviation over three runs; best results are bolded.

| Block | HotpotQA | GSM8K | Avg. |
|---|---|---|---|
| 2 | **75.62$\pm$0.19** | **94.06$\pm$0.11** | **84.84** |
| 1 | 75.43$\pm$0.27 | 93.90$\pm$0.15 | 84.67 |

*Table 14.* Sensitivity to the UCB exploration coefficient $\alpha$. We vary the exploration strength while keeping the validation budget and all other MASPOB components fixed. Results are mean accuracy (%) $\pm$ standard deviation over three runs; best results are bolded.

| $\alpha$ | DROP | MATH | Avg. |
|---|---|---|---|
| 0.05 | 79.27$\pm$0.16 | 55.97$\pm$0.90 | 67.62 |
| 0.2 | 82.28$\pm$0.55 | **57.05$\pm$0.51** | **69.67** |
| 1.0 | **82.38$\pm$0.15** | 55.49$\pm$0.99 | 68.94 |

DROP–MATH setting. This result suggests that diagonal LinUCB offers a practical accuracy–efficiency tradeoff for larger-scale deployments.

### C.4. Block-Coordinate Search under Stronger Coupling

We view standard coordinate ascent as a practical search strategy for our current budgeted setting, especially when combined with UCB exploration. As shown in Table 6, coordinate ascent achieves performance close to exhaustive global search while reducing runtime by 98–99.8%. Intuitively, the UCB bonus assigns larger acquisition values to under-explored prompt combinations, which provides global perturbations that help coordinate ascent avoid poor local optima.

To examine whether single-agent coordinate ascent becomes too restrictive when prompts are more strongly coupled, we evaluate a block-coordinate variant that jointly updates small groups of connected agents. Table 13 shows that block size 2 improves over standard coordinate ascent on HotpotQA and GSM8K. This indicates that local joint updates can better capture upstream–downstream dependencies, while remaining substantially cheaper than exhaustive search over the full prompt tuple. As MAS workflows grow larger, block-coordinate updates over tightly connected agents provide a natural extension, trading higher per-step cost for better modeling of local prompt interactions.

### C.5. Exploration-Coefficient Sensitivity

Finally, we study the sensitivity to the UCB exploration coefficient $\alpha$. Table 14 shows that moderate exploration performs best: insufficient exploration may miss promising prompt regions, whereas overly aggressive exploration can waste the limited evaluation budget on uncertain but low-value combinations. Based on this result, we set $\alpha = 0.2$ in the main experiments.

## D. Prompt Combinations and MAS Workflows for Different Datasets

This section presents the optimal prompt combinations discovered by MASPOB for each benchmark dataset. For every dataset, we provide: (1) the optimized prompts assigned to each agent in the multi-agent system, and (2) the workflow implementation showing how agents collaborate to solve tasks. These configurations were identified through our bandit-guided optimization process, which efficiently explored the combinatorial prompt space while balancing exploration and exploitation.

## D.1. Agent Definitions and Implementation

This subsection provides the implementation details of all agents used in our multi-agent system workflows. Each agent is implemented as an asynchronous operator that interfaces with the language model through a unified API. The `self.prompt` attribute can be dynamically injected by MASPOB during optimization.

---

### AnswerGenerate Agent

**Description:** Generates answers for multi-hop question answering tasks. Used in HotpotQA workflow.

```python
class AnswerGenerate(Operator):
    def __init__(self, llm: AsyncLLM, name: str = "AnswerGenerate"):
        super().__init__(llm, name)
        self.prompt = ANSWER_GENERATION_PROMPT  # Dynamically injected

    async def __call__(self, input: str) -> Dict[str, str]:
        prompt = self.prompt.format(input=input)
        response = await self._fill_node(AnswerGenerateOp, prompt, mode="xml_fill")
        return response
```

---

### ScEnsemble Agent

**Description:** Implements self-consistency ensemble voting. Selects the most frequent answer among multiple candidates. Based on *Self-Consistency Improves Chain of Thought Reasoning in Language Models* (Wang et al., 2023).

```python
class ScEnsemble(Operator):
    def __init__(self, llm: AsyncLLM, name: str = "ScEnsemble"):
        super().__init__(llm, name)
        self.prompt = SC_ENSEMBLE_PROMPT  # Dynamically injected

    async def __call__(self, solutions: List[str], problem: str):
        answer_mapping = {}
        solution_text = ""
        for index, solution in enumerate(solutions):
            answer_mapping[chr(65 + index)] = index
            solution_text += f"{chr(65 + index)}: \n{str(solution)}\n\n\n"

        prompt = self.prompt.format(question=problem, solutions=solution_text)
        response = await self._fill_node(ScEnsembleOp, prompt, mode="xml_fill")

        answer = response.get("solution_letter", "").strip().upper()
        if answer and answer in answer_mapping:
            return {"response": solutions[answer_mapping[answer]]}
        else:
            return {"response": solutions[0] if solutions else ""}
```

---

### FormatAnswer Agent

**Description:** Formats the final answer into a concise response. Used in HotpotQA workflow after ensemble voting.

```python
class FormatAnswer(Operator):
    def __init__(self, llm: AsyncLLM, name: str = "FormatAnswer"):
        super().__init__(llm, name)
        self.prompt = FORMAT_ANSWER_PROMPT  # Dynamically injected

    async def __call__(self, question: str, best_answer: str) -> Dict[str, str]:
        input_text = f"Question: {question}\nBest answer: {best_answer}"
        prompt = self.prompt.format(input=input_text)
        response = await self._fill_node(FormatAnswerOp, prompt, mode="xml_fill")
        return {"answer": response.get("answer", best_answer)}
```

---

### Solve Agent

**Description:** Solves reading comprehension and numerical calculation tasks. Used in DROP and GSM8K workflows.

```python
class Solve(Operator):
    def __init__(self, llm: AsyncLLM, name: str = "Solve"):
        super().__init__(llm, name)
        self.prompt = SOLVE_PROMPT  # Dynamically injected
```

```python
    async def __call__(self, input: str) -> Dict[str, str]:
        prompt = self.prompt.format(input=input)
        response = await self._fill_node(SolveOp, prompt, mode="xml_fill")
        return response
```

## Format Agent

**Description:** Formats the solution into a concise final answer. Used in DROP workflow.

```python
class Format(Operator):
    def __init__(self, llm: AsyncLLM, name: str = "Format"):
        super().__init__(llm, name)
        self.prompt = FORMAT_PROMPT  # Dynamically injected

    async def __call__(self, problem: str, solution: str) -> Dict[str, str]:
        prompt = self.prompt.format(problem_description=problem, solution=solution)
        response = await self._fill_node(FormatOp, prompt, mode="single_fill")
        return response
```

## Programmer Agent

**Description:** Generates and executes Python code to solve mathematical problems. Includes automatic retry with error feedback. Used in GSM8K and MATH workflows.

```python
class Programmer(Operator):
    def __init__(self, llm: AsyncLLM, name: str = "Programmer"):
        super().__init__(llm, name)
        self.prompt = PYTHON_CODE_VERIFIER_PROMPT  # Dynamically injected

    async def __call__(self, problem: str, analysis: str = "None"):
        code, output, feedback = None, None, ""
        for i in range(3):  # Retry up to 3 times
            code_response = await self.code_generate(
                problem, analysis, feedback, mode="code_fill")
            code = code_response.get("code")
            if not code:
                return {"code": code, "output": "No code generated"}
            status, output = await self.exec_code(code)
            if status == "Success":
                return {"code": code, "output": output}
            else:
                feedback = f"\nError from previous attempt:\n{output}"
        return {"code": code, "output": output}
```

## Extract Agent

**Description:** Extracts the final numerical answer from a solution. Used in GSM8K workflow.

```python
class Extract(Operator):
    def __init__(self, llm: AsyncLLM, name: str = "Extract"):
        super().__init__(llm, name)
        self.prompt = GSM8K_EXTRACT_PROMPT  # Dynamically injected

    async def __call__(self, input: str):
        prompt = self.prompt.format(input=input)
        response = await self._fill_node(GenerateOp, prompt, mode="single_fill")
        return response
```

## RefineAnswer Agent

**Description:** Refines and formats mathematical solutions based on code execution output. Produces LaTeX-formatted answers. Used in MATH workflow.

```python
class RefineAnswer(Operator):
    def __init__(self, llm: AsyncLLM, name: str = "RefineAnswer"):
        super().__init__(llm, name)
        self.prompt = REFINE_ANSWER_PROMPT  # Dynamically injected

    async def __call__(self, input: str) -> dict:
```

```
        prompt = self.prompt.format(input=input)
        response = await self._fill_node(RefineAnswerOp, prompt, mode="single_fill")
        return response
```

### GenerateSolution Agent

**Description:** Generates step-by-step mathematical solutions with LaTeX notation. Used in MATH workflow.

```python
class GenerateSolution(Operator):
    def __init__(self, llm: AsyncLLM, name: str = "GenerateSolution"):
        super().__init__(llm, name)
        self.prompt = GENERATE_SOLUTION_PROMPT  # Dynamically injected

    async def __call__(self, input: str) -> dict:
        prompt = self.prompt.format(input=input)
        response = await self._fill_node(GenerateSolutionOp, prompt, mode="single_fill")
        return response
```

### DetailedSolution Agent

**Description:** Generates comprehensive, educational mathematical solutions with detailed explanations. Used in MATH workflow.

```python
class DetailedSolution(Operator):
    def __init__(self, llm: AsyncLLM, name: str = "DetailedSolution"):
        super().__init__(llm, name)
        self.prompt = DETAILED_SOLUTION_PROMPT  # Dynamically injected

    async def __call__(self, input: str) -> dict:
        prompt = self.prompt.format(input=input)
        response = await self._fill_node(DetailedSolutionOp, prompt, mode="single_fill")
        return response
```

### CustomCodeGenerate Agent

**Description:** Generates Python code for programming tasks. Used in HumanEval and MBPP workflows.

```python
class CustomCodeGenerate(Operator):
    def __init__(self, llm: AsyncLLM, name: str = "CustomCodeGenerate"):
        super().__init__(llm, name)
        self.prompt = CUSTOM_CODE_GENERATE_PROMPT  # Dynamically injected

    async def __call__(self, problem: str, entry_point: str):
        full_prompt = self.prompt.format(problem=problem, entry_point=entry_point)
        response = await self._fill_node(
            GenerateOp, full_prompt, mode="code_fill", function_name=entry_point)
        return response
```

### CodeGenerate Agent

**Description:** Generates Python functions for coding problems. Used in MBPP workflow.

```python
class CodeGenerate(Operator):
    def __init__(self, llm: AsyncLLM, name: str = "CodeGenerate"):
        super().__init__(llm, name)
        self.prompt = CODE_GENERATE_PROMPT  # Dynamically injected

    async def __call__(self, problem: str, entry_point: str) -> dict:
        prompt = self.prompt.format(problem=problem, entry_point=entry_point)
        response = await self._fill_node(
            CodeGenerateOp, prompt, mode="code_fill", function_name=entry_point)
        return response
```

**Test Agent**

**Description:** Executes code against test cases and iteratively refines the solution based on error feedback. Used in HumanEval and MBPP workflows.

```python
class Test(Operator):
    def __init__(self, llm: AsyncLLM, name: str = "Test"):
        super().__init__(llm, name)
        self.prompt = REFLECTION_ON_PUBLIC_TEST_PROMPT  # Dynamically injected

    async def __call__(self, problem, solution, entry_point, test_loop: int = 5):
        for _ in range(test_loop):
            result = self.exec_code(solution, entry_point)
            if result == "no error":
                return {"result": True, "solution": solution}
            # Reflect on error and generate fixed solution
            prompt = self.prompt.format(
                problem=problem, solution=solution,
                exec_pass="executed successfully", test_fail=result)
            response = await self._fill_node(ReflectionTestOp, prompt, mode="code_fill")
            solution = response.get("response", solution)
        return {"result": False, "solution": solution}
```

**FixCode Agent**

**Description:** Analyzes error messages and fixes code while preserving the original function signature. Used in MBPP workflow.

```python
class FixCode(Operator):
    def __init__(self, llm: AsyncLLM, name: str = "FixCode"):
        super().__init__(llm, name)
        self.prompt = FIX_CODE_PROMPT  # Dynamically injected

    async def __call__(self, problem: str, solution: str,
                       error: str, entry_point: str) -> dict:
        prompt = self.prompt.format(problem=problem, solution=solution, error=error)
        response = await self._fill_node(
            FixCodeOp, prompt, mode="code_fill", function_name=entry_point)
        return response
```

## D.2. Default Workflow

**MAS Workflow and Optimized Prompt Combination on HotpotQA**

**ANSWER_GENERATION_PROMPT:**

```
Please analyze the question provided in {input}.  Begin by thoroughly examining all details and
identifying the main goal and any sub-goals.  Consider various approaches to the solution, ensuring
to account for negative numbers, zeros, and correct units.  As you reason through the problem,
encapsulate your thought process within <thought> tags.  Once you have reached a conclusion,
present your final answer within <answer> tags.  Ensure your logic is sound and your answer is
accurate before concluding.
```

**SC_ENSEMBLE_PROMPT:**

```
Given the question {question}, analyze the provided solutions: {solutions}.  1.  Identify the
frequency of each answer option.  2.  Consider potential biases or errors in the responses.
3.  Determine which answer appears most frequently among the candidates.  4.  Reflect on
any limitations in the data or possible misinterpretations.  <thought> After reviewing the
frequency of each option, the most common answer is clear.  </thought> <solution_letter> A/B/C
</solution_letter>
```

**FORMAT_ANSWER_PROMPT:**

```
Extract and format a concise final answer from the provided question and best answer.  Ensure
the response is a short phrase, name, or number enclosed in <answer> tags.  Follow these steps:
1.  Carefully analyze the question to determine what it is specifically asking.  2.  Review the
best answer to identify the key information.  3.  Apply logical reasoning to ensure the answer is
accurate.  4.  Confirm that the answer type aligns with the question's requirements.  5.  Present
the final answer succinctly.
```

**Workflow Implementation:**

```python
async def __call__(self, problem: str) -> str:
    # Self-consistency: generate 3 candidate answers
    solutions = []
    for _ in range(3):
        response = await self.answer_generate(input=problem)
        solutions.append(response['answer'])

    # Majority voting ensemble
    ensemble = await self.sc_ensemble(solutions=solutions, problem=problem)
    best_answer = ensemble['response']

    # Format final answer
    result = await self.format_answer(question=problem, best_answer=best_answer)
    return result['answer']
```

The HotpotQA workflow employs three agents. The `AnswerGenerate` agent is executed three times to generate diverse candidate answers through multi-hop reasoning. The `ScEnsemble` agent performs majority voting to select the most frequent answer. Finally, the `FormatAnswer` agent extracts a concise final answer.

---

**MAS Workflow and Optimized Prompt Combination on DROP**

**SOLVE_PROMPT:**

> Read the passage and question carefully. Calculate or identify the specific numerical answer
> requested. Provide a clear step-by-step solution, and ensure your final answer is a single number
> or short phrase without any additional text. For example: If asked "How many years between 1990
> and 2000?", respond with: 1990 to 2000 is a span of: 2000 – 1990 = 10 years. 10

**FORMAT_PROMPT:**

> In response to {problem_description}, provide your answer as {solution}. Ensure it is concise and
> accurate, limited to a short phrase or a few words. Avoid any additional explanations or details.
> Given {problem_description}, your final answer should be {solution}. Keep it brief and to the
> point, with no extra commentary.

---

**Workflow Implementation:**

```python
async def __call__(self, problem: str) -> str:
    # Step 1: Reading comprehension + numerical calculation
    solve_result = await self.solve(input=problem)
    solution = solve_result.get('answer', '')

    # Step 2: Extract concise final answer
    format_result = await self.format(problem=problem, solution=solution)
    return format_result.get('solution', solution)
```

The DROP workflow employs two agents. The `Solve` agent performs reading comprehension and numerical calculation with step-by-step reasoning. The `Format` agent extracts a concise final answer using multiple paraphrased instructions for robust formatting.

---

**MAS Workflow and Optimized Prompt Combination on HumanEval**

**CUSTOM_CODE_GENERATE_PROMPT:**

> Generate a Python function for the HumanEval problem defined by {problem}. The function should
> be named {entry_point} and should adhere to the following requirements: 1. Clearly define the
> inputs and outputs based on the problem statement. 2. Ensure the logic is implemented step by
> step, considering edge cases. 3. Write clean, readable code that follows Python best practices.
> Your final output should be a complete function definition that can be tested independently.

**SC_ENSEMBLE_PROMPT:**

> You are tasked with determining the most frequent answer from a list of candidates. Follow these
> steps carefully: 1. Analyze the provided {solutions} to identify how many times each option
> appears. 2. Consider the implications of selecting the most frequent answer. What if there is
> a tie? 3. Ensure that your selection process is based solely on the frequency of the answers
> provided in {solutions}, without making any assumptions. 4. After analyzing, clearly mark your
> thought process within <thought> tags and your final answer with <solution_letter> tags.

**REFLECTION_ON_PUBLIC_TEST_PROMPT:**

```
    Given a code problem and a python code solution which failed to pass test or execute, you need to
    carefully analyze the reason for the failure and propose a corrected code solution.  1.  Understand
    the Expected Behavior:  Re-read the problem description carefully.  Pay special attention to edge
    cases mentioned in examples.  2.  Analyze the Test Failure:  Compare the expected output with
    actual output from the failed test case.  3.  Identify the Root Cause:  Common issues include
    misunderstanding requirements, off-by-one errors, incorrect handling of negative numbers.  4.  Fix
    the Code:  Make the minimal necessary changes.  Provide ONLY the corrected Python code solution.
```

**Workflow Implementation:**

```python
async def __call__(self, problem: str, entry_point: str) -> str:
    # Step 1: Generate 3 candidate solutions
    solutions = []
    for _ in range(3):
        solution = await self.custom_code_generate(
            problem=problem, entry_point=entry_point)
        if solution.get('response'):
            solutions.append(solution['response'])

    # Step 2: Ensemble to select best solution
    if len(solutions) >= 2:
        best = await self.sc_ensemble(solutions=solutions, problem=problem)
        best_code = best.get('response', solutions[0])
    else:
        best_code = solutions[0]

    # Step 3: Test and fix if needed
    test_result = await self.test(
        problem=problem, solution=best_code,
        entry_point=entry_point, test_loop=5)
    return test_result.get('solution', best_code)
```

The HumanEval workflow employs three agents. The `CustomCodeGenerate` agent is executed three times to generate diverse code solutions with step-by-step implementation. The `ScEnsemble` agent performs majority voting to select the most consistent solution among candidates. Finally, the `Test` agent executes the code against public test cases and iteratively refines the solution if failures occur.

---

**MAS Workflow and Optimized Prompt Combination on MBPP**

**CODE_GENERATE_PROMPT:**

```
    Generate a Python function to solve the given problem.  Ensure the function name matches the one
    specified in the problem.  Include necessary imports.  Use clear variable names and add comments
    for clarity.  Problem: {problem}.  Function signature: {entry_point}.  Generate the complete
    function below:
```

**SC_ENSEMBLE_PROMPT:**

```
    Given the question {question}, analyze the provided solutions:  {solutions}.  1.  Identify the
    frequency of each answer option.  2.  Consider potential biases or errors in the responses.
    3.  Determine which answer appears most frequently among the candidates.  4.  Reflect on
    any limitations in the data or possible misinterpretations.  <thought> After reviewing the
    frequency of each option, the most common answer is clear.  </thought> <solution_letter> A/B/C
    </solution_letter>
```

**REFLECTION_ON_PUBLIC_TEST_PROMPT:**

```
    Given a code problem and a python code solution which failed to pass test or execute, you need to
    carefully analyze the reason for the failure and propose a corrected code solution.  1.  Understand
    the Expected Behavior:  Re-read the problem description carefully.  2.  Analyze the Test Failure:
    Compare the expected output with actual output.  3.  Identify the Root Cause:  Common issues
    include off-by-one errors, incorrect handling of negative numbers.  4.  Fix the Code:  Make the
    minimal necessary changes.  Provide ONLY the corrected Python code solution.
```

**FIX_CODE_PROMPT:**

```
    Analyze the provided code to identify the reason for the test failures.  Begin by examining the
    {error} reported in the test logs.  Consider the logic within the function, ensuring that all edge
    cases are handled, especially potential issues like division by zero or overflow.  1.  Review the
    relevant sections of the code where the {problem} may arise.  2.  Test the function with a variety
    of inputs to replicate the failures.  3.  Once the root cause is established, devise a {solution}
    that corrects the issue while keeping the function signature unchanged.
```

---

**Workflow Implementation:**

```python
async def __call__(self, problem: str, entry_point: str) -> str:
    # Step 1: Generate 3 candidate solutions in parallel
    results = await asyncio.gather(
        self.code_generate(problem=problem, entry_point=entry_point),
        self.code_generate(problem=problem, entry_point=entry_point),
        self.code_generate(problem=problem, entry_point=entry_point))
    candidate_codes = [r.get('code') for r in results if r.get('code')]

    # Step 2: Ensemble to select best solution
    if len(candidate_codes) >= 2:
        ensemble = await self.sc_ensemble(solutions=candidate_codes, problem=problem)
        best_code = ensemble.get('response', candidate_codes[0])
    else:
        best_code = candidate_codes[0]

    # Step 3: Test the code
    test_result = await self.test(
        problem=problem, solution=best_code, entry_point=entry_point)

    if test_result.get('result'):
        return test_result.get('solution', best_code)

    # Step 4: Fix code if test failed
    fix_result = await self.fix_code(
        problem=problem, solution=best_code, error=str(test_result))
    return fix_result.get('code', best_code)
```

The MBPP workflow employs four agents. The `CodeGenerate` agent is executed three times in parallel to generate diverse code solutions. The `ScEnsemble` agent performs majority voting to select the most consistent solution. The `Test` agent executes the code against public test cases and identifies failures. Finally, the `FixCode` agent analyzes error messages and proposes corrections while preserving the original function signature.

---

**MAS Workflow and Optimized Prompt Combination on GSM8K**

**GSM8K_SOLVE_PROMPT:**

```
Solve this math problem step by step.  Show all your work clearly and end with a numerical answer.
Break down the solution into:  1.  Given information 2.  Step-by-step calculations 3.  Final
numerical answer clearly marked with ** **.  Make sure to:  Include all mathematical operations,
Show intermediate calculations, Double check your arithmetic, Consider all given values in the
problem, Verify your answer makes logical sense.  Problem:  {input}
```

**SC_ENSEMBLE_PROMPT:**

```
Given the question {question}, you will analyze the provided solutions {solutions} to determine
which answer appears most frequently among the candidates.  1.  Begin by identifying each solution
and counting how many times each one appears.  2.  Consider the context of each solution and its
relevance to the question.  3.  Document your thought process in <thought> tags, explaining how
you arrived at your conclusion.  4.  After thorough analysis, select the most frequent answer and
denote it with <solution_letter> tags.
```

**PYTHON_CODE_VERIFIER_PROMPT:**

```
Write a Python function named 'solve' that addresses the following mathematical problem:
{problem}.  Begin by breaking down the problem into smaller parts, ensuring you understand each
component clearly.  Conduct a thorough {analysis} of the requirements and any potential edge cases.
After developing the logic, implement the function while maintaining clarity and efficiency in the
code.  Finally, review your work to ensure it meets the problem's criteria.
```

**GSM8K_EXTRACT_PROMPT:**

```
Extract numerical answer from {input}.  Look for:  ** ** markers, Programmer solution comparison,
Final calculated value.  Return:  single number.
```

---

**Workflow Implementation:**

```python
async def __call__(self, problem: str) -> str:
    # Step 1: Generate 3 candidate solutions
    solutions = []
    for _ in range(3):
        solution = await self.solve(input=problem)
        solutions.append(solution.get('response', ''))

    # Step 2: Ensemble to select best solution
```

```
    if len(solutions) >= 2:
        ensemble = await self.sc_ensemble(solutions=solutions, problem=problem)
        best_solution = ensemble.get('response', solutions[0])
    else:
        best_solution = solutions[0]

    # Step 3: Verify with Programmer (code execution)
    prog_result = await self.programmer(problem=problem, analysis=best_solution)
    prog_output = prog_result.get('output', '')

    # Step 4: Extract final numerical answer
    extract_input = best_solution + "\nProgrammer solution: " + prog_output
    final_result = await self.extract(input=extract_input)
    return final_result.get('response', '')
```

The GSM8K workflow employs four agents in a sequential pipeline. The `Solve` agent is executed three times to generate diverse step-by-step solutions with clearly marked numerical answers. The `ScEnsemble` agent performs majority voting to select the most consistent solution. The `Programmer` agent then verifies the calculation by executing Python code. Finally, the `Extract` agent parses the solution to return only the final numerical answer.

---

**MAS Workflow and Optimized Prompt Combination on MATH**

**PYTHON_CODE_VERIFIER_PROMPT:**

    Write a Python function named 'solve' that addresses the following mathematical problem:
    {problem}. Ensure that your solution is efficient and handles potential edge cases. Before
    finalizing your code, perform a thorough {analysis} of the problem to identify any possible errors
    or assumptions. After coding, provide {feedback} on the solution's correctness and efficiency.

**REFINE_ANSWER_PROMPT:**

    Given the mathematical problem and the output from the code execution, please provide a
    well-formatted and detailed solution. Follow these guidelines: 1. Begin with a clear statement
    of the problem. 2. Explain the approach and any formulas or concepts used. 3. Show step-by-step
    calculations, using LaTeX notation for mathematical expressions. 4. Interpret the code output
    and incorporate it into your explanation. 5. Provide a final answer, enclosed in \boxed{} LaTeX
    notation.

**DETAILED_SOLUTION_PROMPT:**

    Provide a comprehensive, educational solution to the following mathematical problem: {input}.
    Begin by analyzing what is being asked and identifying the given information. Evaluate the problem
    systematically, using detailed explanations and LaTeX notation where appropriate. State the answer
    directly and concisely. Ensure to double-check all calculations and logic, looking for potential
    off-by-one errors. Consider multiple approaches to the solution and verify that the final answer
    makes sense in the context of the problem.

**GENERATE_SOLUTION_PROMPT:**

    Please solve the given mathematical problem step by step. Follow these guidelines: 1. State
    the problem clearly. 2. Outline the approach and any relevant formulas or concepts. 3. Provide
    detailed calculations, using LaTeX notation for mathematical expressions. 4. Explain each step of
    your reasoning. 5. Present the final answer enclosed in \boxed{} LaTeX notation. Your solution
    should be thorough, mathematically sound, and easy to understand.

**SC_ENSEMBLE_PROMPT:**

    Given the following question and multiple answer candidates, your task is to determine the most
    frequent answer among the provided solutions. Analyze the candidates carefully, considering their
    frequencies and any patterns that may emerge. Pay attention to any ties and ensure you select the
    most frequent option based on your analysis. <thought> First, I will count the occurrences of each
    answer in the solutions provided. Then, I will identify which answer appears the most frequently.
    </thought> <solution_letter> A/B/C </solution_letter>

---

**Workflow Implementation:**

```
async def __call__(self, problem: str) -> str:
    # Parallel execution of 4 branches
    async def branch_programmer_refine():
        code_result = await self.programmer(problem=problem)
        code_output = code_result.get('output', '')
        if code_output and 'Error' not in str(code_output):
            refine_input = f"{problem}\nCode output:\n{code_output}"
            refined = await self.refine_answer(input=refine_input)
```

```
            return refined.get('response')
        return None

    async def branch_detailed():
        detailed = await self.detailed_solution(input=problem)
        return detailed.get('response')

    async def branch_generate():
        generated = await self.generate_solution(input=problem)
        return generated.get('response')

    # Execute all branches in parallel
    results = await asyncio.gather(
        branch_programmer_refine(),  # Branch 1: Code + Refine
        branch_detailed(),           # Branch 2: Detailed solution
        branch_generate(),           # Branch 3: Generate solution
        branch_generate())           # Branch 4: Generate solution

    solutions = [r for r in results if r]

    # Ensemble voting for final answer
    ensemble = await self.sc_ensemble(solutions=solutions, problem=problem)
    return ensemble.get('response', solutions[0])
```

The MATH workflow employs five agents organized in a parallel branching structure. The `Programmer` agent generates Python code to solve the problem, and its output is passed to the `RefineAnswer` agent for formatting with LaTeX notation. Concurrently, the `DetailedSolution` agent provides a comprehensive educational solution, while the `GenerateSolution` agent is executed twice to generate additional step-by-step solutions. Finally, the `ScEnsemble` agent performs majority voting across all four branches to select the most consistent final answer.

### D.3. Complex Workflow on HotpotQA, DROP, and HumanEval

---

**MAS Workflow and Optimized Prompt Combination on HotpotQA (Complex)**

**ANSWER_GENERATION_PROMPT:**

```
Using logical reasoning, please answer the following question: {input}.  <thought> Analyze the
question carefully to understand what is being asked.  Consider any boundary conditions and ensure
that the answer type matches the question.  Apply the appropriate method to arrive at a conclusion.
Test your answer mentally to confirm its accuracy.  </thought> <answer> Provide your final answer
here.  </answer>
```

**HOTPOTQA_EXTRACT_DIRECT_PROMPT:**

```
Extract the key entities or facts from the following question without any additional context.  Your
response should focus solely on the essential information requested.  Here is the input:  {input}
Identify the main entities or facts directly from this question:  {input} Please analyze the
question carefully and provide only the relevant entities or facts.  Input to consider:  {input}
From the question provided, extract the specific entities or facts without any surrounding context.
Use the following input:  {input} Determine the critical entities or facts in this question.
Respond only with the extracted information based on the input:  {input}
```

**HOTPOTQA_EXTRACT_CONTEXT_PROMPT:**

```
Using the provided information, your task is to extract relevant entities or facts by applying
reasoning to the context given in {input}.  Carefully analyze the context to identify key details,
relationships, and implications.  Consider the following steps:  Begin by thoroughly reading
{input} to grasp the main idea and supporting details.  Identify the key entities and facts that
are mentioned, ensuring you understand their significance within the context.  Evaluate how these
entities are interconnected and what reasoning can be derived from their relationships.  Reflect
on any limitations or boundaries that may affect your understanding of the context.  Ensure that
your final answer is not only accurate but also reasonable in magnitude and aligns with similar
problems you've encountered.  After considering all these aspects, provide your answer based on the
reasoning you've developed.  The answer is {reasoning}.
```

**HOTPOTQA_VERIFY_PROMPT:**

```
Verify if the proposed answer {answer} is consistent with the context {context} and accurately
addresses the question {question}.  If it does, confirm it as verified; if not, provide the
corrected answer.  Ensure logical reasoning is applied throughout the verification process.
```

**HOTPOTQA_TERMINOLOGY_PROMPT:**

---

To accurately refine your answer using precise terminology, follow these steps: 1. Begin by thoroughly understanding the {context} provided. Identify key concepts and terms related to the {question}. 2. List what you already know about the topic and pinpoint any gaps in your knowledge that need to be addressed. 3. Ensure you check for any potential off-by-one errors that may arise in your calculations or reasoning. 4. Assess the magnitude of your answer to ensure it is reasonable in relation to the {context}. 5. Use relevant examples to clarify your thought process and guide your selection of terminology. 6. After considering all aspects, determine the most accurate term that aligns with the {context} and directly answers the {question}. Finally, confirm your findings and state: 'The answer is {answer}'.

**HOTPOTQA_CROSS_REFERENCE_PROMPT:**

Examine the provided answers thoroughly and cross-reference them to ensure completeness and accuracy. Follow these steps: 1. Identify the key components of {question} and the information presented in {context_answer} and {current_answer}. 2. Analyze the reasoning behind each answer, paying close attention to any edge cases or special conditions that may affect the overall completeness. 3. Evaluate the validity of each answer by substituting back into the context, ensuring that they align with the given information. 4. Consider multiple approaches to the solution and determine which answer provides the most comprehensive response to the question. After thorough examination and cross-referencing, output the most complete answer on a new line. {reasoning} {direct_answer}

**SC_ENSEMBLE_PROMPT:**

Given the question {question}, analyze the provided solutions: {solutions}. 1. Identify the frequency of each answer option. 2. Consider potential biases or errors in the responses. 3. Determine which answer appears most frequently among the candidates. 4. Reflect on any limitations in the data or possible misinterpretations. <thought> After reviewing the frequency of each option, the most common answer is clear. I must ensure that I have accurately counted each occurrence and that no answer has been overlooked. </thought> <solution_letter> A/B/C </solution_letter>

**FORMAT_ANSWER_PROMPT:**

Extract and format a concise final answer from the provided question and best answer. The answer should be presented in <answer> tags. Ensure that you consider all relevant aspects, check for edge cases, and verify the accuracy of the information. Your response should be direct and clear, focusing on the key requirements. Use the following format: 1. Analyze the question and the best answer given in {input}. 2. Identify the essential information and any special conditions. 3. Formulate a short phrase, name, or number that encapsulates the answer. 4. Present your final answer in <answer> tags. Remember to double-check your answer for clarity and correctness before finalizing.

**Workflow Implementation:**

```python
async def __call__(self, problem: str) -> str:
    # Stage 1: Initial answer generation
    answer_result = await self.generate(input=problem)
    initial_answer = answer_result.get('answer', '')
    reasoning = answer_result.get('thought', '')

    # Stage 2-3: Parallel entity extraction (two branches)
    async def branch_direct():
        direct = await self.extract_direct(input=problem)
        return direct.get('response')

    async def branch_context():
        context = await self.extract_context(input=problem, reasoning=reasoning)
        return context.get('response')

    direct_answer, context_answer = await asyncio.gather(
        branch_direct(),     # Branch 1: Direct extraction
        branch_context())    # Branch 2: Context-based extraction

    # Stage 4: Answer verification
    verified = await self.verify(question=problem, answer=initial_answer,
                                 context=f"{direct_answer}\n{context_answer}")
    verified_answer = verified.get('response', initial_answer)

    # Stage 5: Terminology refinement
    refined = await self.terminology(question=problem, answer=verified_answer,
                                     context=reasoning)
    refined_answer = refined.get('response', verified_answer)

    # Stage 6: Cross-reference verification
    cross_ref = await self.cross_reference(question=problem,
                                           current_answer=refined_answer,
                                           direct_answer=direct_answer,
                                           context_answer=context_answer,
```

```
                                              reasoning=reasoning)
    cross_ref_answer = cross_ref.get('response', refined_answer)

    # Stage 7: Ensemble voting
    solutions = [verified_answer, refined_answer, cross_ref_answer]
    ensemble = await self.sc_ensemble(question=problem, solutions=solutions)
    best_answer = ensemble.get('response', cross_ref_answer)

    # Stage 8: Final answer formatting
    final = await self.format_answer(input=f"{problem}\n{best_answer}")
    return final.get('answer', best_answer)
```

This complex workflow for HotpotQA implements an 8-stage multi-hop question answering pipeline with verification and refinement. The `AnswerGenerate` agent first produces an initial answer with reasoning. Two parallel extraction branches then process the question: `HotpotQAExtractDirect` extracts entities directly from the question, while `HotpotQAExtractContext` uses the reasoning context to identify relevant facts. The `HotpotQAVerify` agent validates the answer against both extraction results. Subsequently, `HotpotQATerminology` refines the answer using precise terminology, and `HotpotQACrossReference` performs comprehensive cross-referencing to ensure completeness. The `ScEnsemble` agent applies majority voting across multiple answer candidates, and finally, `FormatAnswer` produces a concise, well-formatted final response.

---

### MAS Workflow and Optimized Prompt Combination on DROP (Complex)

**DROP_QA_PROMPT:**

```
Instruct the model to extract numerical values from provided text passages with a focus on
accuracy. Ensure that the model answers questions directly without elaboration or additional
commentary. Thoroughly analyze the passage for details. Identify all relevant numerical
information, including negative numbers and zeros. Confirm the accuracy of the extracted values
and ensure units are correctly stated. Consider different methods to arrive at the answer if
applicable. Provide the final answer on a separate line. Answer:
```

**DROP_NUMERICAL_PROMPT:**

```
Verify the numerical accuracy of the following calculations by identifying all numbers present
and validating each calculation step. Carefully analyze the context and ensure all mathematical
operations are correct. Cross-check your findings with alternative methods if necessary. Pay
special attention to any potential errors or misinterpretations. Once you have completed the
verification process, provide only the final verified numerical answer. Your task is to ensure
that every number in the following statement is accurate. Validate the calculations thoroughly.
Be meticulous in your approach and ensure no detail is overlooked. After completing your checks,
output the verified numerical answer only. Begin by examining the numerical components of the
given data. Identify all instances of numbers and perform the necessary calculations to confirm
their accuracy. Consider different methods to cross-verify your results. Once you have thoroughly
validated the calculations, present only the final numerical answer. Your objective is to
scrutinize the following information for numerical correctness. Locate all numbers, validate the
calculations, and ensure they align with mathematical principles. Take your time to cross-check
your results and handle any special cases with care. At the end of your analysis, provide solely
the verified numerical answer.
```

**DROP_CALC_PROMPT:**

```
Perform precise mathematical validation for the following scenarios: 1. Calculate the percentage
of X out of Y. 2. Determine the time taken if the speed is Z and the distance is W. 3. Find the
quantity of items if the total cost is A and the price per item is B. Provide only the calculated
number for each scenario.
```

**DROP_VERIFY_PROMPT:**

```
Please verify the final answer and format it correctly. Convert any word numbers into digits and
ensure that the answer adheres to proper formatting standards. Output only the corrected answer.
```

**ANSWER_GENERATION_PROMPT:**

```
You are tasked with answering a question. Begin by comprehensively understanding {input}. Break
down the problem into manageable steps, carefully considering each aspect. Use <thought> tags
to articulate your reasoning process clearly. Ensure you are thorough in your analysis, avoiding
common pitfalls and double-checking any critical calculations. Once you have worked through the
problem, present your final answer using <answer> tags. Make sure your response is organized
and concise. 1. Read and comprehend {input}. 2. Identify key components and separate the
problem into steps. 3. Analyze each step thoroughly, placing your reasoning in <thought> tags.
4. Conclude with your final answer in <answer> tags. Remember to review your reasoning before
finalizing your answer.
```

**FORMAT_PROMPT:**

```
In order to provide a precise and succinct response to {problem_description}, please ensure that
your final answer is a brief phrase, name, or a few words.  Follow these steps:  Begin by fully
understanding the question.  Identify the key information you have and what you need to determine.
Carefully verify your calculations and logic to avoid any errors.  Make sure the magnitude of your
answer is reasonable and check for any off-by-one mistakes.  Use relevant examples to clarify your
thought process.  Once you have completed these steps, format your answer concisely.  The answer is
{solution}.
```

**SC_ENSEMBLE_PROMPT:**

```
You are tasked with determining the most frequent answer from a list of candidates.  Follow
these steps carefully: 1.  Analyze the provided {solutions} to identify how many times each
option appears.  2.  Consider the implications of selecting the most frequent answer.  What if
there is a tie?  3.  Ensure that your selection process is based solely on the frequency of the
answers provided in {solutions}, without making any assumptions.  4.  After analyzing, clearly
mark your thought process within <thought> tags and your final answer with <solution_letter>
tags.  Remember to thoroughly parse the {question} and verify your findings by substituting back
into the context.  Eliminate any options that do not meet the frequency criteria.  Your final
response should only include the answer in the specified format.  <thought>After analyzing the
frequency of each option in {solutions}, I find that option A appears most frequently.</thought>
<solution_letter>A</solution_letter>
```

**Workflow Implementation:**

```python
async def __call__(self, problem: str, passage: str) -> str:
    # Stage 1: Question Analysis and Initial Extraction
    qa_result = await self.drop_qa(passage=passage, question=problem)
    initial_answer = qa_result.get('response', '')

    # Stage 2: Numerical Accuracy Verification
    numerical_result = await self.drop_numerical(
        answer=initial_answer, passage=passage)
    verified_numbers = numerical_result.get('response', initial_answer)

    # Stage 3: Mathematical Calculation Validation
    calc_result = await self.drop_calc(
        answer=verified_numbers, passage=passage)
    calculated_answer = calc_result.get('response', verified_numbers)

    # Stage 4: Answer Verification and Formatting
    verify_result = await self.drop_verify(answer=calculated_answer)
    verified_answer = verify_result.get('response', calculated_answer)

    # Stage 5: Answer Generation with Reasoning
    generate_result = await self.answer_generate(
        input=f"{passage}\n\nQuestion: {problem}\nCandidate: {verified_answer}")
    generated_answer = generate_result.get('answer', verified_answer)

    # Stage 6: Final Formatting
    format_result = await self.format(
        problem_description=problem, solution=generated_answer)
    formatted_answer = format_result.get('response', generated_answer)

    # Stage 7: Ensemble Voting
    solutions = [verified_answer, generated_answer, formatted_answer]
    ensemble = await self.sc_ensemble(question=problem, solutions=solutions)
    return ensemble.get('response', formatted_answer)
```

## MAS Workflow and Optimized Prompt Combination on DROP (Complex)

**DROP_QA_PROMPT:**

```
Instruct the model to extract numerical values from provided text passages with a focus on
accuracy.  Ensure that the model answers questions directly without elaboration or additional
commentary.  Thoroughly analyze the passage for details.  Identify all relevant numerical
information, including negative numbers and zeros.  Confirm the accuracy of the extracted values
and ensure units are correctly stated.  Consider different methods to arrive at the answer if
applicable.  Provide the final answer on a separate line.  Answer:
```

**DROP_NUMERICAL_PROMPT:**

```
Verify the numerical accuracy of the following calculations by identifying all numbers present
and validating each calculation step.  Carefully analyze the context and ensure all mathematical
operations are correct.  Cross-check your findings with alternative methods if necessary.  Pay
special attention to any potential errors or misinterpretations.  Once you have completed the
```

verification process, provide only the final verified numerical answer. Your task is to ensure that every number in the following statement is accurate. Validate the calculations thoroughly. Be meticulous in your approach and ensure no detail is overlooked. After completing your checks, output the verified numerical answer only. Begin by examining the numerical components of the given data. Identify all instances of numbers and perform the necessary calculations to confirm their accuracy. Consider different methods to cross-verify your results. Once you have thoroughly validated the calculations, present only the final numerical answer. Your objective is to scrutinize the following information for numerical correctness. Locate all numbers, validate the calculations, and ensure they align with mathematical principles. Take your time to cross-check your results and handle any special cases with care. At the end of your analysis, provide solely the verified numerical answer.

### DROP_CALC_PROMPT:

Perform precise mathematical validation for the following scenarios: 1. Calculate the percentage of X out of Y. 2. Determine the time taken if the speed is Z and the distance is W. 3. Find the quantity of items if the total cost is A and the price per item is B. Provide only the calculated number for each scenario.

### DROP_VERIFY_PROMPT:

Please verify the final answer and format it correctly. Convert any word numbers into digits and ensure that the answer adheres to proper formatting standards. Output only the corrected answer.

### ANSWER_GENERATION_PROMPT:

You are tasked with answering a question. Begin by comprehensively understanding {input}. Break down the problem into manageable steps, carefully considering each aspect. Use <thought> tags to articulate your reasoning process clearly. Ensure you are thorough in your analysis, avoiding common pitfalls and double-checking any critical calculations. Once you have worked through the problem, present your final answer using <answer> tags. Make sure your response is organized and concise. 1. Read and comprehend {input}. 2. Identify key components and separate the problem into steps. 3. Analyze each step thoroughly, placing your reasoning in <thought> tags. 4. Conclude with your final answer in <answer> tags. Remember to review your reasoning before finalizing your answer.

### FORMAT_PROMPT:

In order to provide a precise and succinct response to {problem_description}, please ensure that your final answer is a brief phrase, name, or a few words. Follow these steps: Begin by fully understanding the question. Identify the key information you have and what you need to determine. Carefully verify your calculations and logic to avoid any errors. Make sure the magnitude of your answer is reasonable and check for any off-by-one mistakes. Use relevant examples to clarify your thought process. Once you have completed these steps, format your answer concisely. The answer is {solution}.

### SC_ENSEMBLE_PROMPT:

You are tasked with determining the most frequent answer from a list of candidates. Follow these steps carefully: 1. Analyze the provided {solutions} to identify how many times each option appears. 2. Consider the implications of selecting the most frequent answer. What if there is a tie? 3. Ensure that your selection process is based solely on the frequency of the answers provided in {solutions}, without making any assumptions. 4. After analyzing, clearly mark your thought process within <thought> tags and your final answer with <solution_letter> tags. Remember to thoroughly parse the {question} and verify your findings by substituting back into the context. Eliminate any options that do not meet the frequency criteria. Your final response should only include the answer in the specified format. <thought>After analyzing the frequency of each option in {solutions}, I find that option A appears most frequently.</thought> <solution_letter>A</solution_letter>

**Workflow Implementation:**

```python
async def __call__(self, problem: str, passage: str) -> str:
    # Stage 1: Question Analysis and Initial Extraction
    qa_result = await self.drop_qa(passage=passage, question=problem)
    initial_answer = qa_result.get('response', '')

    # Stage 2: Numerical Accuracy Verification
    numerical_result = await self.drop_numerical(
        answer=initial_answer, passage=passage)
    verified_numbers = numerical_result.get('response', initial_answer)

    # Stage 3: Mathematical Calculation Validation
    calc_result = await self.drop_calc(
        answer=verified_numbers, passage=passage)
    calculated_answer = calc_result.get('response', verified_numbers)

    # Stage 4: Answer Verification and Formatting
    verify_result = await self.drop_verify(answer=calculated_answer)
    verified_answer = verify_result.get('response', calculated_answer)
```

```python
    # Stage 5: Answer Generation with Reasoning
    generate_result = await self.answer_generate(
        input=f"{passage}\n\nQuestion: {problem}\nCandidate: {verified_answer}")
    generated_answer = generate_result.get('answer', verified_answer)

    # Stage 6: Final Formatting
    format_result = await self.format(
        problem_description=problem, solution=generated_answer)
    formatted_answer = format_result.get('response', generated_answer)

    # Stage 7: Ensemble Voting
    solutions = [verified_answer, generated_answer, formatted_answer]
    ensemble = await self.sc_ensemble(question=problem, solutions=solutions)
    return ensemble.get('response', formatted_answer)
```

This complex workflow for DROP implements a 7-stage numerical question answering pipeline with multi-level verification. The `DropQA` agent first extracts numerical values from the passage with a focus on accuracy. The `DropNumerical` agent then validates all numbers and calculations by cross-checking with alternative methods. The `DropCalc` agent performs precise mathematical validation including percentage, time, and quantity calculations. The `DropVerify` agent ensures proper formatting by converting word numbers to digits. Subsequently, the `AnswerGenerate` agent produces a comprehensive answer with reasoning, followed by the `Format` agent which condenses the response to a concise phrase. Finally, the `ScEnsemble` agent performs majority voting across multiple answer candidates to select the most reliable final answer.

---

**MAS Workflow and Optimized Prompt Combination on HumanEval (Complex)**

**CUSTOM_CODE_GENERATE_PROMPT:**

Generate a Python function for the following problem: {problem}. The function should be defined with the name {entry_point}. Ensure it adheres to best practices and is optimized for performance. Include necessary input validation and edge case handling.

**HUMANEVAL_ANALYZE_PATTERN_PROMPT:**

Analyze the following Python code for proper algorithmic patterns and data structures. Consider the key requirements and constraints of the problem. Check for edge cases and special conditions that might affect the validity of the implementation. After thorough evaluation, determine if the code adheres to best practices in algorithm design and data structure usage. {code} Output 'valid' or 'invalid' only.

**HUMANEVAL_ANALYZE_COMPLEXITY_PROMPT:**

Analyze the following Python code for its time and space complexity. Determine if it is optimal. Output 'valid' if the complexities are optimal, otherwise output 'invalid'. {code}

**HUMANEVAL_VALIDATE_CODE_PROMPT:**

Validate the following code snippet: {code}. Determine if it is clean Python with valid syntax and contains a single function. Respond with 'valid' or 'invalid' only. Assess the code provided: {code}. Check for clarity, proper syntax, and ensure it defines only one function. Output 'valid' if it meets these criteria, otherwise output 'invalid'. Examine this code: {code}. Verify if it adheres to Python's syntax rules and consists of a single function. Your response should be either 'valid' or 'invalid'. Review the code snippet: {code}. Confirm whether it is syntactically correct, clean, and contains only one function. Provide 'valid' or 'invalid' as your answer.

**HUMANEVAL_VERIFY_SOLUTION_PROMPT:**

To determine the validity of the solution provided, follow these steps: 1. Analyze the {problem} to identify the specific requirements and expected output format. 2. Examine the {solution} to ensure it meets all outlined criteria. 3. Check for consistency between the {solution} and the expected results from the {problem}. 4. Consider edge cases and limits to ensure robustness. 5. Confirm that all outputs align with the specified format. After completing these evaluations, conclude with either 'valid' or 'invalid' based on your findings. Your answer is:

**SC_ENSEMBLE_PROMPT:**

Given the question {question}, you are tasked with analyzing a set of possible answers provided in {solutions}. Your goal is to determine which answer appears most frequently among the candidates. 1. Start by carefully examining each candidate answer within {solutions}. 2. Count the occurrences of each answer option. 3. Analyze the results and identify the one that is the most frequent. 4. Use <thought> tags to document your reasoning and thought process throughout this analysis. 5. Once you have determined the most frequent answer, indicate it with <solution_letter> tags, using the corresponding letter (A/B/C) based on the options provided. Be methodical in your approach and ensure that you review your findings before presenting your final answer. Answer:

**REFLECTION_ON_PUBLIC_TEST_PROMPT:**

Analyze the following code failure in tests. Identify the reasons behind the failure and propose a solution. Ensure to consider edge cases and logical errors. Break down the problem into manageable parts, and provide a clear and concise explanation. Your response should be enclosed in <reflection_and_solution> tags. {exec_pass}: [Insert execution pass details here] {test_fail}: [Insert test failure details here] {problem}: [Describe the problem causing the test failure] {solution}: [Provide a solution to fix the identified problem] <reflection_and_solution>

**Workflow Implementation:**

```python
async def __call__(self, problem: str, entry_point: str) -> str:
    # Stage 1: Generate Initial Code Solution
    code_result = await self.custom_code_generate(
        problem=problem, entry_point=entry_point)
    generated_code = code_result.get('response', '')

    # Stage 2: Analyze Algorithmic Patterns
    pattern_result = await self.analyze_pattern(code=generated_code)
    pattern_valid = pattern_result.get('response', '') == 'valid'

    # Stage 3: Analyze Time/Space Complexity
    complexity_result = await self.analyze_complexity(code=generated_code)
    complexity_valid = complexity_result.get('response', '') == 'valid'

    # Stage 4: Validate Code Syntax and Structure
    validate_result = await self.validate_code(code=generated_code)
    syntax_valid = validate_result.get('response', '') == 'valid'

    # Stage 5: Verify Solution Against Problem Requirements
    verify_result = await self.verify_solution(
        problem=problem, solution=generated_code)
    solution_valid = verify_result.get('response', '') == 'valid'

    # Stage 6: Ensemble Voting on Multiple Code Candidates
    candidates = [generated_code]  # Can include multiple generated versions
    ensemble = await self.sc_ensemble(question=problem, solutions=candidates)
    best_code = ensemble.get('response', generated_code)

    # Stage 7: Test Reflection and Refinement
    test_result = await self.test(
        problem=problem, solution=best_code,
        exec_pass="", test_fail="")
    final_code = test_result.get('solution', best_code)

    return final_code
```

This complex workflow for HumanEval implements a 7-stage code generation and verification pipeline with multi-level validation. The `CustomCodeGenerate` agent first produces an optimized Python function with proper input validation and edge case handling. The `HumanEvalAnalyzePattern` agent then evaluates whether the code demonstrates appropriate algorithmic patterns and data structure usage. Next, the `HumanEvalAnalyzeComplexity` agent assesses optimal time and space complexity. The `HumanEvalValidateCode` agent verifies clean Python syntax and single-function structure. Subsequently, the `HumanEvalVerifySolution` agent confirms that the solution meets all problem requirements and expected output formats. The `ScEnsemble` agent performs majority voting across multiple code candidates to select the most reliable implementation. Finally, the `Test` agent reflects on any test failures and proposes fixes by analyzing edge cases and logical errors.

