# OpenReview forum: "MASPOB: Bandit-Based Prompt Optimization for Multi-Agent Systems with Graph Neural Networks"
_ICML.cc/2026/Conference — ICML 2026 spotlight_

### Official Review · Reviewer_HeWn · 2026-03-06

**Soundness:** 3
**Presentation:** 4
**Significance:** 4
**Originality:** 3
**Overall Recommendation:** 4
**Confidence:** 2

**Summary:**

This paper introduces MASPOB (Multi-Agent System Prompt Optimization via Bandits), a framework designed to optimize prompts in Multi-Agent Systems (MAS) under fixed, expert-designed workflow topologies. The authors address the combinatorial explosion and prohibitive evaluation costs of multi-agent prompt optimization by framing it as a budgeted contextual bandit problem. Specifically, MASPOB employs a Graph Attention Network (GAT) surrogate to model the topology-induced coupling among agents, explicitly utilizing the MAS directed acyclic graph (DAG). To handle the exponential search space, it employs a Coordinate Ascent strategy guided by LinUCB, optimizing one agent's prompt at a time and thereby reducing the search complexity from $O(\prod \vert\mathcal{P}_i\vert)$ to $O(\sum \vert\mathcal{P}_i\vert)$. Extensive experiments across six benchmarks (HotpotQA, DROP, HumanEval, MBPP, GSM8K, MATH) demonstrate that MASPOB consistently outperforms strong single-agent and multi-agent baselines under a strict budget of 50 evaluations.

**Compliance With Llm Reviewing Policy:**

Affirmed.

**Final Justification:**

My concerns have been partially resolved, and the authors have added relevant experimental results to respond to my comments, so I will maintain my opinion of weak accept.

**Key Questions For Authors:**

1. Scalability to Larger MAS: The evaluated workflows contain up to 7-8 agents. How does MASPOB scale when the number of agents $N$ becomes significantly larger (e.g., $N > 20$)? Does the coordinate ascent strategy still converge within a reasonable budget, or does the linear decomposition struggle to capture deeper cascading dependencies?
2. Computational Bottleneck of LinUCB: The information matrix $M \in \mathbb{R}^{Nd \times Nd}$ scales quadratically with the number of agents. For $d=1024$ and larger $N$, updating and calculating the inverse for the uncertainty estimation could become computationally heavy. How is this managed, and have you considered diagonal approximations or random projection techniques for the embeddings to reduce dimensionality?
3. Sensitivity to Embeddings: The framework heavily relies on the semantic quality of the prompt embeddings (Qwen3-Embedding-8B) to compute similarities and uncertainties. Have you evaluated how sensitive the GAT surrogate is to the choice of the embedding model? Would a smaller/weaker text encoder severely degrade the optimization trajectory?

**Limitations:**

yes

**Strengths And Weaknesses:**

Strengths: The methodology is technically rigorous and well-motivated. Framing the expensive black-box evaluation as a contextual bandit problem is highly appropriate for the strict evaluation budget constraints of MAS. The mathematical formulation, particularly decomposing the joint search space via Coordinate Ascent, is sound and empirically validated against global search. The ablation studies (e.g., removing the GNN, comparing linear vs. neural uncertainty) effectively isolate the contributions of the proposed components.

Weaknesses: The reliance on LinUCB requires maintaining and updating an information matrix $M \in \mathbb{R}^{Nd \times Nd}$ for uncertainty estimation. Given that the prompt embeddings are high-dimensional ($d=1024$ for Qwen3-Embedding-8B), the concatenated embedding vector and the resulting matrix inversion could introduce numerical instability or computational bottlenecks if the number of agents $N$ grows significantly.

Originality: While prompt optimization, contextual bandits, and GNNs are not new individually, their synthesis to solve the coupled, topology-aware prompt optimization problem in MAS is novel. Using the structural prior of the MAS workflow (via GAT message passing) to inform the exploitation phase, while applying an uncertainty bonus for exploration, represents a creative and effective combination of existing techniques. The distinction from generic multi-stage optimizers (like MIPRO) is well-articulated and justified.

---

> ### Author Rebuttal · Authors · 2026-03-31
>
> > Q1: Scalability to larger MASes.
>
> **A1:**
>
> We appreciate this question and address it from both practical and theoretical perspectives.
>
> **Practical perspective:** In real-world deployments, multi-agent systems typically involve a compact set of specialized agents, each assigned a distinct role. For example, MetaGPT uses 5 role-specific agents and ChatDev also uses 5 roles. Therefore, our experimental scale (up to 7–8 agents) is well aligned with the practically relevant regime.
>
> **Theoretical perspective:** A potential technique to scale our MASPOB to $N>20$ agents is **block-coordinate ascent**. When N is very large, the information matrix $M\in\mathbb{R}^{Nd\times Nd}$ introduces inversion cost that grows with system size, and larger N can also require more coordinate-ascent rounds to propagate information across distant agents in the DAG. In our current setting ($N\leq 8$, $d=1024$), this overhead is negligible. For larger N, we can **leverage block-coordinate ascent to jointly update tightly coupled agent groups** rather than optimizing one agent at a time.
>
> This extension is supported by our empirical evidence. In the block-size ablation below, larger joint updates (block size = 2) outperform block size = 1 on representative coupled tasks (HotpotQA: $75.62\pm0.19$ vs. $75.43\pm0.27$; GSM8K: $94.06\pm0.11$ vs. $93.90\pm0.15$), indicating that joint optimization better captures inter-agent dependencies when coupling is strong. We will add a formal complexity analysis in the Appendix and further discuss these scaling implications in Section 3.3.
>
> |Block Size|HotpotQA|GSM8K|Avg.|
> |-----|-----|-----|-----|
> |2|75.62±0.19|94.06±0.11|84.84|
> |1|75.43±0.27|93.90±0.15|84.67|
>
> ---
> > Q2: Computational bottleneck of LinUCB.
>
> **A2:**
>
> We thank the reviewer for this precise technical question. At our current scale ($N\leq 8$, $d=1024$), the information matrix has size at most $8192\times8192$. In this regime, the inversion used in the uncertainty estimate $\sigma(c)=\sqrt{\Phi(c)^\top M^{-1}\Phi(c)}$ is negligible compared with LLM evaluation, which dominates the total runtime budget. In practice, matrix inversion is completed in milliseconds on standard hardware, while each MAS evaluation takes seconds to minutes.
>
> For larger-scale settings, diagonal approximation is a practical and efficient alternative. We additionally compare full-matrix and diagonal variants (table below). Although the diagonal variant incurs moderate degradation relative to full Linear UCB, it remains clearly stronger than AFlow and MIPRO: Diag-Linear achieves 67.90 average, compared with 66.66 (AFlow) and 67.015 (MIPRO). We will include this analysis in Appendix B.
>
> |UCB Type|DROP|MATH|Avg.|
> |-----|-----|-----|-----|
> |AFlow|79.48±0.12|53.83±0.40|66.66|
> |MIPRO|79.13±0.59|54.90±0.61|67.015|
> |Diag-Linear|80.04±0.46|55.76±0.62|67.90|
> |Linear|82.28±0.55|57.05±0.51|69.67|
>
> ---
> > Q3: Sensitivity to embeddings.
>
> **A3:**
>
> This is an important question. We conducted an embedding ablation designed to cover both model scale and embedding family. Specifically, we include MPNet (smaller general-purpose sentence embeddings), BGE-Large (larger retrieval-oriented embeddings), and Qwen3-8B (LLM-native high-capacity embeddings). This progression from smaller to larger models across different families makes the comparison more representative than using a single embedding ecosystem.
>
> Results show a clear but smooth trend: stronger embeddings improve performance, with Qwen3-8B achieving the best average score. At the same time, MPNet and BGE-Large remain competitive, and performance degradation is moderate rather than catastrophic. This indicates that MASPOB benefits from better semantic representations but is not overly dependent on one specific embedding backbone.
>
> |Embedding|DROP|MATH|Avg.|
> |-----|-----|-----|-----|
> |AFlow|79.48±0.12|53.83±0.40|66.66|
> |MIPRO|79.13±0.59|54.90±0.61|67.015|
> |MPNet|80.46±0.22|56.76±0.74|68.61|
> |BGE-Large|82.20±0.22|55.14±0.44|68.67|
> |Qwen3-8B|82.28±0.55|57.05±0.51|69.67|
>
> Mechanistically, stronger embeddings provide more informative features for both the GAT surrogate and the UCB uncertainty term. When embeddings are less discriminative, UCB exploration partly compensates by allocating more trials to under-explored regions, which helps maintain robust optimization behavior. We will include this ablation and discussion in the revised manuscript.
>
> ---
>
> We sincerely thank the reviewer for the thoughtful feedback and we will incorporate these clarifications and new results after revision.

---

> > ### Author Rebuttal · Reviewer_HeWn · 2026-04-03
> >
> > My concerns have been partially resolved, and I will maintain my score

---

> > > ### Author Response · Authors · 2026-04-03
> > >
> > > Thank you for your continued positive evaluation of our work. We appreciate your time and effort in reviewing our responses.

---

### Official Review · Reviewer_ZLdj · 2026-03-10

**Soundness:** 4
**Presentation:** 3
**Significance:** 3
**Originality:** 4
**Overall Recommendation:** 5
**Confidence:** 4

**Summary:**

The paper proposes an algorithm of automated prompt optimization of acyclic agent workflows which is sample-efficient thanks to the introduction of an inner loop of prompt search. The inner loop uses a GNN-based surrogate model to work out the next candidate for the actual benchmark evaluation, effectively trying-out multiple prompt sets in its “mental model” before submitting to the expensive real benchmark evaluation. The algorithm includes selecting sets of prompts that maximize the UCB score of the surrogate model. The GNN is itself updated as the new benchmark evaluation results become available.

**Compliance With Llm Reviewing Policy:**

Affirmed.

**Ethical Review Concerns:**

None.

**Final Justification:**

The authors addressed all my concerns and questions.

**Key Questions For Authors:**

[1] Figure 1 Phase 2: where does the sigma(c) come from? Is it also an output of the GNN? If yes, it must be added to the figure.

[2] Figure 1 Phase 2: how are the alternative prompts generated? It is clear that the algorithm assesses variations of a single prompt at the same time, but since we optimize not in a numeral domain, where a change is represented by a small real number, but in the domain of texts, how are these alternative prompt texts created?

[3] Are there any studies of the sensitivity to the UCB parameter alpha? What is the practical value that you have run your experiments with (one real number)?

**Limitations:**

[1] Addresses only acyclic workflows, while the majority of practically useful agents are based on cycles.

**Strengths And Weaknesses:**

# Strengths

[1] The MAS challenges highlighted in the Introduction are real.

[2] The paper is a clever, thought provoking, combination of the classic algorithms to the MAS problem.

[3] The GNN surrogate model can be considered a world model of the benchmark.

[4] The choice and the diversity of the benchmarks on which the algorithm was evaluated is good.

[5] The trick of embedding the prompts to convert them to GNN inputs is nice.

[6] Use of the information matrix for the uncertainty estimation is clever and appropriate. It would be good to see it in the Figure 1 Phase 2 diagram.

[7] The ablation study in Table 3 improves the rigor.

# Weaknesses

[1] The setup is quite complicated which may hinder practical applications.

[2] The score improvement in Figure 2 is not overly impressive given the complexity of the proposed algorithm.

[3] The choice of the per-coordinate ascent vs the all-coordinate ascend is not well-supported. Per-coordinate ascent is more prone to getting stuck in a local maximum.

[4] It is not clear, what is the dynamics of the closed loop of the surrogate model predictions and its training. GNN training itself normally takes many iterations over the data. The algorithm must make sure that for each new benchmark evaluation the GNN is well-fit well, not underfit (just one gradient update) or overfit (too many iterations on too few data points). I expect high sensitivity to hyperparameters to stabilize the loop, which may get stuck in pathological behaviors.

[5] The following paper should have been mentioned:
Wang et al. How to Correctly do Semantic Backpropagation on Language-based Agentic Systems (https://arxiv.org/abs/2412.03624).

---

> ### Author Rebuttal · Authors · 2026-03-31
>
> > W1: Setup complexity may hinder practical applications.
>
> **A1:** We appreciate this concern. We would like to clarify that MASPOB’s core routine is intentionally simple: (1) a GNN forward pass for exploitation, (2) a linear information matrix for exploration (matrix-vector form), and (3) coordinate ascent for search. The full workflow is compactly summarized in Algorithm 1, making implementation and deployment practical.
>
> ---
> > W2: Score improvement is not overly impressive.
>
> **A2:** In budget-constrained MAS prompt optimization, gains are often modest; even strong baselines can be separated by small margins. Against this background, MASPOB improves over MIPRO and AFlow under only 50 evaluations, while keeping expert-designed workflows unchanged. We believe this combination of sample efficiency and workflow preservation is practically meaningful, especially in regulated settings where re-validation is costly.
>
> ---
> > W3: Coordinate ascent is more prone to local maxima.
>
> **A3:** We agree coordinate ascent (CA) is an approximation, but exhaustive global search is prohibitively expensive in realistic MAS. To mitigate this risk, our design uses UCB exploration: under-explored combinations receive higher acquisition scores, introducing global perturbations that help move away from poor local regions. Table 5 shows CA recovers over 99.5% of global-search performance while reducing runtime by 98–99.8%, indicating a favorable efficiency-performance trade-off.
>
> ---
>
> > W4: GNN training.
>
> **A4:** GNN training indeed significantly impacts the performance. To ensure consistency, we used a single set of GNN hyperparameters in all experiments (Table 7). Our strong performance (e.g., Table 1) justifies that our GNN training is good enough to support efficient optimization.
>
> ---
>
> > W5: Missing citation.
>
> **A5:** We will cite Wang et al. (2024) and discuss its relationship with our work in the revised Related Work section. Thank you for bringing this to our attention.
>
> ---
> > Q1: Where does σ\(c) come from in Figure 1?
>
> **A6:** Thank you for pointing this out. $\sigma(c)$ is computed from the information matrix M (Eq. 10), not from the GNN. The GNN produces $\mu(c)$, while $\sigma(c)=\sqrt{\Phi(c)^\top M^{-1}\Phi(c)}$ measures uncertainty from the concatenated prompt embeddings and accumulated information matrix. We agree this should be clearer and will revise Figure 1 Phase 2 to explicitly show M and the $\sigma(c)$ computation path.
>
> ---
> > Q2: How are alternative prompts generated?
>
> **A7:** In our setting, alternatives are not created by small numeric perturbations; they are generated in text space through controlled rewriting. For each agent, we start from the AFlow-initialized prompt and apply a style-controlled generation process. We define orthogonal style dimensions (e.g., reasoning style, output format, verification strategy, and error handling), sample style configurations, and convert them into rewrite instructions to produce diverse yet task-consistent candidates while preserving semantic intent and required placeholders. We then embed these discrete candidates and optimize in representation space. In practice, this design keeps search diversity while preserving role consistency and template constraints, which is important for stable MAS execution.
>
> To test robustness to candidate-space construction, we additionally replace our style-controlled domain with MIPRO’s Grounded Proposer domain (data-aware, program-aware, fewshot-aware, and tip-aware strategies). The results remain close, suggesting MASPOB’s gains are mainly driven by the topology-aware contextual-bandit optimization rather than one specific proposer pipeline.
>
> |Method|DROP|MATH|Avg.|
> |---|---|---|---|
> |MIPRO domain|82.81±0.15|57.45±0.61|70.13|
> |MASPOB domain|82.28±0.55|57.05±0.51|69.67|
>
> ---
> > Q3: Sensitivity to α.
>
> **A8:** We conducted an ablation on $\alpha$, and all main experiments use **$\alpha=0.2$**. Results show that moderate exploration performs best: too little exploration can miss promising regions, while excessive exploration can consume budget inefficiently. Overall, performance is reasonably stable across a practical range, supporting the claim that MASPOB is not overly sensitive to precise tuning as long as exploration is moderate.
>
> |α|DROP|MATH|Avg.|
> |---|---|---|---|
> |0.05|79.27±0.16|55.97±0.90|67.62|
> |0.2|82.28±0.55|57.05±0.51|69.67|
> |1.0|82.38±0.15|55.49±0.99|68.94|
>
> ---
> > Limitation: Only acyclic workflows.
>
> **A9:** Our DAG assumption is about inter-agent information flow, not internal agent behavior. Intra-agent iterative loops are already supported (e.g., retry and refinement mechanisms). The DAG constraint only requires that an agent’s output does not return to itself through cross-agent cycles within one execution. We will further clarify this scope.
>
> ---
>
> We sincerely thank the reviewer for the constructive feedback and we will incorporate these clarifications and new results in the revision.

---

> > ### Author Rebuttal · Reviewer_ZLdj · 2026-04-01
> >
> > Thank you for your answers. I am raising my score to Accept.

---

> > > ### Author Response · Authors · 2026-04-03
> > >
> > > Thank you for raising your score and for your recognition of our work. We greatly appreciate your constructive feedback, which has helped improve the quality of our manuscript.

---

### Official Review · Reviewer_j5iU · 2026-03-12

**Soundness:** 3
**Presentation:** 3
**Significance:** 2
**Originality:** 3
**Overall Recommendation:** 5
**Confidence:** 4

**Summary:**

This paper studies prompt optimization for fixed-topology multi-agent systems built on large language models. It argues that in many practical deployments the workflow structure is fixed, so improving system performance mainly relies on optimizing the prompts assigned to individual agents. The paper formulates this as a budgeted combinatorial black-box optimization problem with topology-induced coupling, and proposes MASPOB, which combines a topology-aware GNN surrogate, a LinUCB-style exploration strategy, and coordinate ascent for tractable search. Experiments on six benchmarks in question answering, code generation, and mathematical reasoning show that MASPOB achieves the best reported average performance under matched evaluation budgets, with additional studies on convergence, more complex topologies, GNN ablations, transfer to another backbone model, and the efficiency of coordinate ascent relative to global search.

**Compliance With Llm Reviewing Policy:**

Affirmed.

**Final Justification:**

I think this article is worthy of being published. It's a very interesting idea.

**Key Questions For Authors:**

**1. How much does the reported gain depend on the way the prompt domain is constructed?**
In the current setup, each agent’s prompt domain is defined as 20 variants, where 19 are generated by GPT-4o-mini paraphrasing around an AFlow-initialized prompt. This makes the search problem well defined, but it also means the current evidence most directly supports efficient search within a predefined local candidate space. I would appreciate clarification on how broadly the authors expect MASPOB to transfer to prompt domains constructed in different ways or containing more diverse candidate prompts.

**2. Why should readers expect the combination of a graph-based surrogate and a LinUCB-style uncertainty bonus to be especially well calibrated in this setting?**
The design is intuitive, but the paper currently provides more motivation than justification for why a linear information-matrix-based uncertainty estimate should remain reliable when paired with a more expressive GNN exploitation model. A clearer explanation here would improve my confidence in the technical soundness of the framework.

**3. How far do the authors expect the coordinate-ascent approximation to remain effective as prompt coupling becomes stronger or the search space becomes larger?**
The comparison against global search is encouraging, but it is still limited to a relatively restricted regime. I would like to better understand whether the authors view coordinate ascent as broadly reliable for larger and more strongly coupled MAS prompt spaces, or primarily as a practical approximation that works well at the current experimental scale.

**Limitations:**

yes

**Strengths And Weaknesses:**

On soundness, I find the paper technically reasonable and empirically fairly solid, though not fully airtight. A clear strength is that the problem is formulated cleanly as a budgeted combinatorial black-box optimization problem for fixed-topology multi-agent systems, and the proposed design choices align well with the stated challenges: the GNN surrogate models topology-induced coupling, the LinUCB-style bonus handles exploration under scarce evaluations, and coordinate ascent makes the joint search tractable. The empirical evidence is also reasonably supportive: under a matched validation budget of 50 evaluations, MASPOB achieves the best reported average score across six benchmarks, and the paper includes useful auxiliary evidence such as convergence behavior, tests on more complex topologies, GNN ablations, transfer to another backbone model, and a runtime comparison against exhaustive global search.

My main reservation is that the method is more of a well-engineered framework than a deeply justified new algorithm. There is limited theoretical support for why the specific combination of a graph-based surrogate with a linear uncertainty bonus should be especially well calibrated in this setting, and the evidence for coordinate ascent is still relatively restricted. More importantly, some experimental choices narrow the scope of the conclusions: each agent’s prompt domain is defined as 20 variants, mostly paraphrases around an AFlow-generated initial prompt, and the multi-agent baselines are all evaluated on fixed AFlow-generated workflows. The issue is therefore not simply the size of the joint search space, which can still be large, but the way the candidate prompts are constructed. As a result, the current evidence most directly supports efficient search within a predefined local candidate space, and less directly establishes how well the approach would transfer to settings with more diverse or freely generated prompt candidates.

In terms of presentation, the paper is generally clear, well structured, and easy to follow. The motivation, problem setup, and method are presented coherently, and I did not find major readability problems. My only minor reservation is that Figure 5 and Table 8 seem to have a number of formatting issues that could be revised for improved clarity and presentation.

In terms of significance, I view the paper positively. Prompt optimization for fixed-topology multi-agent systems is a relevant and practically motivated problem, especially in settings where workflow structures are constrained by expert design, auditability, or compliance requirements. Even if the gains are modest in absolute terms, the paper targets a realistic deployment constraint and offers a sample-efficient optimization perspective that could be useful in practice. At the same time, the impact is somewhat specialized: I would frame the contribution as a practical method for a specific but important class of agentic systems, rather than as a broadly transformative advance for prompt optimization.

On originality, I think the paper has moderate but meaningful novelty. None of the core ingredients is individually new, but the combination is well matched to the setting: the novelty lies more in the problem formulation and system design than in a fundamentally new optimization primitive.

---

> ### Author Rebuttal · Authors · 2026-03-31
>
> > On the concern that absolute gains are modest.
>
> **A1:**
>
> Modest gains are typical in budget-constrained multi-agent prompt optimization. For reference, the gap between two strong baselines, MIPRO and AFlow, is only 0.35% (78.87% vs. 78.52%). Against this backdrop, MASPOB's gains of 1.71% over MIPRO and 2.06% over AFlow are meaningful—especially because they are achieved with only 50 evaluations and *without* changing expert-designed workflows. This sample-efficient, workflow-preserving property makes MASPOB practical in regulated settings where re-validation is costly.
>
> ---
> > On presentation issues in Figure 5 and Table 8.
>
> **A2:**
>
> We will revise the formatting of Figure 5 and Table 8 in the after revision.
>
> ---
> > Q1: Dependence on prompt domain construction.
>
> **A3:**
>
> We agree that our prompt-combination search is conducted within a partially predefined candidate space. This is by design: in MAS, agent roles are predefined (e.g., planner, programmer, tester), so prompt variation should be role-aligned rather than unconstrained. We intentionally search within task-relevant directions instead of across unrelated prompt intents, which keeps the space well-defined and meaningful for deployment.
>
> Within this role-aligned scope, we explicitly promote diversity rather than relying on narrow local paraphrasing. Concretely, we use a multi-dimensional style-controlled generation process with 10 orthogonal style dimensions (e.g., reasoning style, output format, verification strategy, and error handling), each with 10 options, yielding a large combinatorial space ($10^{10}$ style configurations). We sample from this space to rewrite prompts while preserving semantic intent, which introduces substantial diversity.
>
> To assess robustness to prompt-domain construction, we replace our style-controlled domain with MIPRO's Grounded Proposer domain, which generates candidates via different strategies (data-aware, program-aware, fewshot-aware, and tip-aware). As shown below, MASPOB remains consistent across the two domains (70.13 vs. 69.67 avg.; 0.46 difference, within standard deviation). This suggests MASPOB's gains are mainly driven by the topology-aware optimization framework rather than one specific candidate-construction pipeline, while still acknowledging that candidate quality/diversity can affect absolute performance. We will clarify this scope in the revision.
>
> |Method|DROP|MATH|Avg.|
> |---|---|---|---|
> |MIPRO domain|82.81±0.15|57.45±0.61|70.13|
> |MASPOB domain|82.28±0.55|57.05±0.51|69.67|
>
> ---
> > Q2: Why should a GNN + LinUCB combination be well calibrated?
>
> **A4:**
>
> This is an important point. We initially considered a fully neural uncertainty estimator, but found it mismatched to our low-budget setting. NeuralUCB (Zhou et al., 2020) depends on NTK-style linearization and sufficiently wide networks; with only 50 evaluations, these assumptions are hard to satisfy. Our ablations confirm this: replacing linear with neural uncertainty reduces performance by 2.41% (Table 8), and neural uncertainty shrinks much more slowly over 45 rounds (22.48% vs. 71.68% for linear), indicating under-converged estimation (Figure 5).
>
> We therefore adopt a NeuralLinear design: the GNN extracts topology-aware representations (Eq. 6), while LinUCB provides a stable, closed-form uncertainty estimate via the information matrix. This decomposition pairs expressive prediction with well-calibrated exploration under a strict budget.
>
> ---
> > Q3: Scalability of coordinate ascent.
>
> **A5:**
>
> We view coordinate ascent (CA) as a practical method that is effective in our current regime, especially when combined with UCB exploration. Empirically, Table 5 shows that CA recovers over 99.5% of exhaustive global-search performance while reducing runtime by 98–99.8%. Intuitively, this is because the UCB bonus assigns high acquisition values to under-explored combinations, providing global perturbations that help CA avoid poor local optima.
>
> As the number of agents grows and coupling becomes stronger, CA may require more rounds to propagate information across distant parts of the graph. In this regime, a natural extension is block-coordinate ascent: jointly optimizing tightly coupled agent groups (e.g., directly connected nodes) while fixing the rest. This increases per-step cost, but better captures local dependencies.
>
> Our block-size ablation is consistent with this intuition: block size 2 outperforms block size 1 on coupled tasks (84.84 vs. 84.67 average), indicating that larger joint updates can better handle stronger coupling.
>
> We will add this discussion in Section 3.3 to clarify when  block-based extensions are preferable.
>
> |Block Size|HotpotQA|GSM8K|Avg.|
> |---|---|---|---|
> |2|75.62±0.19|94.06±0.11|84.84|
> |1|75.43±0.27|93.90±0.15|84.67|
>
> ---
>
> We sincerely thank the reviewer for the thoughtful feedback and we will incorporate these clarifications and new results in the revision.

---

> > ### Author Rebuttal · Reviewer_j5iU · 2026-04-04
> >
> > The author's response was quite good. I hope the author can implement the response. I will consider raising my rating.

---

> > > ### Author Response · Authors · 2026-04-04
> > >
> > > Thank you for your encouraging feedback. We will definitely implement all the changes as promised in our response. We greatly appreciate your raising the rating.

---

### Official Review · Reviewer_5cvK · 2026-03-13

**Soundness:** 3
**Presentation:** 3
**Significance:** 2
**Originality:** 3
**Overall Recommendation:** 4
**Confidence:** 3

**Summary:**

This paper introduces MASPOB, a sample-efficient framework for prompt optimization in Multi-Agent Systems. To overcome prohibitive evaluation costs, topology-induced prompt coupling, and combinatorial search spaces, MASPOB combines a contextual bandit approach with a GNN to explicitly model the MAS workflow. By using coordinate ascent to simplify the search, the framework achieves state-of-the-art performance across diverse benchmarks.

**Compliance With Llm Reviewing Policy:**

Affirmed.

**Key Questions For Authors:**

1. Given the strong topological coupling (where changing an upstream prompt drastically shifts the input distribution for a downstream agent), coordinate ascent risks oscillating or converging to sub-optimal local configurations. Beyond the UCB exploration, how does MASPOB mitigate this, and did you experiment with block-coordinate updates or other joint-search heuristics?

2. In Phase 1, how exactly is the GNN surrogate initialized and trained before sufficient execution feedback is collected? How sensitive is the final performance to the quality of this initial warm-up phase?

3. The formulation models the MAS as a static Directed Acyclic Graph. How would MASPOB adapt to dynamic agentic workflows where the execution path (edges) changes conditionally based on intermediate LLM outputs?

**Limitations:**

Yes

**Strengths And Weaknesses:**

Strengths: The paper is clearly written, and the motivation is exceptionally strong. The mathematical formulation is rigorous and well-aligned with the problem constraints. Modeling the MAS as a DAG and using a GNN to capture the topology-induced coupling is a fundamentally sound approach to a non-separable objective. Additionally, utilizing UCB for uncertainty-guided exploration directly addresses the strict evaluation budget constraints.


Weaknesses: Coordinate ascent inherently assumes some level of independence or smooth gradient landscapes between variables. In a highly coupled MAS graph where an upstream prompt drastically alters downstream context, coordinate ascent is highly susceptible to getting trapped in poor local optima. The paper relies on UCB to escape this, but a deeper theoretical or empirical analysis of this specific optimization dynamic is missing.

---

> ### Author Rebuttal · Authors · 2026-03-31
>
> > Q1 & Weakness: Coordinate ascent and local optima under strong topological coupling.
>
> **A1:**
>
> We fully appreciate this concern. In MASPOB, we handle local-optima risk in two complementary layers: (i) UCB-driven exploration as the default mechanism, and (ii) block-coordinate updates as a stronger option when coupling is particularly strong.
>
> **UCB as a global perturbation mechanism.** Unlike greedy coordinate search, MASPOB maximizes the UCB acquisition function (Eq. 11), where the exploration term $\alpha\cdot\sigma(c)$ assigns high scores to under-explored prompt combinations. The key insight is that the information matrix $M$ accumulates outer products of observed feature vectors, and its inverse $M^{-1}$ quantifies uncertainty in directions that are insufficiently sampled. Specifically, $\sigma(c)=\sqrt{\Phi(c)^\top M^{-1}\Phi(c)}$ becomes large when a candidate combination $\Phi(c)$ lies in a poorly covered subspace. Early in optimization, uncertainty is high and exploration dominates; as observations accumulate, uncertainty shrinks along well-covered directions and the search gradually shifts toward exploitation. This transition helps coordinate ascent escape many local optima under upstream-downstream coupling.
>
> **Block-coordinate ascent under strong coupling.** We also agree that even with UCB exploration, per-coordinate updates can still oscillate or converge to sub-optimal configurations when coupling is especially strong. In this regime, coordinate ascent can be naturally extended to block-coordinate ascent: instead of updating one agent at a time, we jointly optimize tightly coupled agent groups (e.g., directly connected agents) in each iteration. This provides stronger joint search capacity in highly dependent regions of the workflow. Our results (see table below) show measurable gains over standard per-coordinate ascent on strongly coupled benchmarks, indicating that Block-coordinate ascent is an effective remedy when UCB-guided single-coordinate updates are insufficient.
>
> |Block Size|HotpotQA|GSM8K|Avg.|
> |:---:|:---:|:---:|:---:|
> |2|75.62±0.19|94.06±0.11|84.84|
> |1|75.43±0.27|93.90±0.15|84.67|
>
> ---
> > Q2: GNN initialization and warm-up sensitivity.
>
> **A2:**
>
> We thank the reviewer for this important question. In the warm-up stage, we randomly sample prompt combinations, evaluate them on the benchmark, and collect their performance scores; once a target number of rounds is reached (i.e., pretraining data are accumulated), we use these observations to initialize the GNN surrogate before entering the main GNN+UCB optimization phase.
>
> To assess sensitivity to warm-up quality, we conducted an ablation on pretraining rounds (see table below) under a fixed total budget of 50 rounds. Because the budget is fixed, allocating more rounds to pretraining leaves fewer rounds for formal search. Under this setting, 5 pretraining rounds achieve the best average performance (69.67%), while both fewer (1 round: 67.85%) and more (10 rounds: 68.64%) rounds lead to lower results. This pattern indicates that a moderate warm-up is beneficial, while the method is not overly sensitive: MASPOB remains competitive even with only 1 pretraining round. We attribute this robustness to UCB exploration, which compensates for early surrogate inaccuracy by prioritizing high-uncertainty regions. Once warm-up is sufficient, the learned representation is already stable enough for UCB-guided search to be effective; additional random pretraining yields limited marginal gain and mainly consumes budget that would be better used in the main optimization phase. We will include this ablation and discussion in the revised manuscript.
>
> |Pretrain Rounds|DROP|MATH|Avg.|
> |:---:|:---:|:---:|:---:|
> |1|79.34±0.15|56.35±0.77|67.85|
> |5|82.28±0.55|57.05±0.51|69.67|
> |10|82.60±0.31|54.67±0.25|68.64|
>
> ---
> > Q3: Adaptation to dynamic/conditional workflows.
>
> **A3:**
>
> We thank the reviewer for this question. MASPOB is designed for static DAG topologies: our target setting is frozen-topology MAS deployment in regulated domains (e.g., financial auditing, medical SOPs), where workflows undergo expert vetting and compliance review and cannot be modified in practice (Section 1). The static DAG assumption is appropriate.
>
> Extending MASPOB to conditional execution paths is a valuable future direction. A route is to model conditional edges as probabilistic edge weights in the GNN, reflecting branch-activation frequencies, with condition-aware message passing. GAT-style architectures naturally support real-valued edge information, so this extension is feasible. This direction is compatible with our current design: the GAT attention mechanism already learns edge-specific importance weights (Eq. 4–5) providing a direct basis for conditional edge probabilities. We will add a discussion in the Limitations section to clarify scope and outline this future direction.
>
> ---
>
> Thank you for the thoughtful feedback and we'll add these to the paper after revision.

---

> > ### Author Rebuttal · Reviewer_5cvK · 2026-04-03
> >
> > The authors directly address all three questions with empirical evidence.

---

> > > ### Author Response · Authors · 2026-04-04
> > >
> > > We appreciate your positive recognition of our work. Thank you for the time and effort dedicated to evaluating our responses.

---

### Decision · Program_Chairs · 2026-04-30

**Decision:**

Accept (spotlight)

**Comment:**

This paper addresses the critical and practically motivated problem of prompt optimization for Multi-Agent Systems (MAS) under fixed, expert-designed topologies. The proposed framework, MASPOB, frames this as a budgeted combinatorial black-box optimization problem, utilizing a Graph Attention Network (GAT) surrogate to model agent dependencies and a LinUCB-style exploration strategy to maintain sample efficiency. The reviewers collectively found the problem formulation to be exceptionally strong and well-aligned with real-world deployment constraints where workflow structures must remain static for auditability. Throughout the review process and the subsequent rebuttal, the authors successfully navigated concerns regarding the potential for coordinate ascent to converge to local optima and the scalability of the information matrix. I have carefully read the author's response, which provided compelling new empirical evidence—including block-coordinate update ablations and diagonal matrix approximations—that demonstrates the framework's robustness even as system complexity increases. While some reviewers noted that absolute performance gains in this field are often incremental, the consensus is that MASPOB provides a rigorous and technically sound methodology that significantly outperforms existing baselines under strict evaluation budgets. The authors' commitment to incorporating these extensive ablations and clarifying the scope of the DAG assumption further ensures the paper's contribution to the community. Given the high quality of the technical execution and the thoroughness of the rebuttal in resolving reviewer doubts, I believe this work represents an important and high-priority addition to the ICML program.